

# One loop verification of SMEFT Ward Identities

**Tyler Corbett$^\star$ and Michael Trott$^2$**

Niels Bohr Institute, Copenhagen, Denmark

$\star$ corbett.t.s@gmail.com

## Abstract

We verify Standard Model Effective Field Theory Ward identities to one loop order when background field gauge is used to quantize the theory. The results we present lay the foundation of next to leading order automatic generation of results in the SMEFT, in both the perturbative and non-perturbative expansion using the geoSMEFT formalism, and background field gauge.

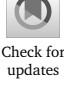

# 1  Introduction

The Standard Model Effective Field Theory (SMEFT) [1, 2] is a core theory for interpreting many current and future experimental measurements in particle physics. The SMEFT is defined by the field content of the Standard Model, including an $\mathrm{SU_L}(2)$ scalar Higgs doublet ($H$), and a linear realization of $\mathrm{SU(3)} \times \mathrm{SU_L}(2) \times \mathrm{U(1)_Y}$ symmetry. Operators of mass dimension $d$ are suppressed by powers of an unknown non-Standard Model scale $\Lambda^{d-4}$.

The SM treated as an EFT has both derivative and field expansions. The Higgs field expansion plays an essential role as it can collapse terms in a composite operator onto a target n-point interaction when the classical background field expectation value of the Higgs is taken. This introduces modifications of low n-point functions, and the corresponding Lagrangian parameters such as the masses, gauge couplings and mixing angles. These modifications result in much of the interesting phenomenology of the SMEFT.

Actively organising the formulation of the SMEFT using field space geometry is advantageous. This approach is known as the geoSMEFT [3], and builds on the theoretical foundation laid down in Refs. [4–10]. The geoSMEFT separates out the scalar field space expansion (in a gauge independent manner) from the derivative expansion. This approach naturally generalizes the SM Lagrangian parameters to their SMEFT counterparts, which are understood to be the masses, gauge couplings and mixing angles on the curved background Higgs manifold.[1] The degree of curvature of the Higgs field spaces is dictated by the ratio of the Electroweak scale $\bar{v}_T \equiv \sqrt{\langle 2H^\dagger H \rangle}$ compared to the scale of new physics $\Lambda$. The geoSMEFT enables all orders results in the $\bar{v}_T/\Lambda$ expansion to be defined, due to the constraints of a self consistent description of the geometry present in the theory, and has already resulted in the first exact formulation of the SMEFT to $\mathcal{O}(\bar{v}_T^4/\Lambda^4)$ [11].

Organizing the SMEFT using field space geometry can be done while background field gauge invariance is maintained by using the Background Field Method (BFM). The BFM is also advantageous, as then gauge fixing does not obscure naive and intuitive one loop Ward-Takahashi identities [12, 13] (hereafter referred to as Ward identities for brevity) that reflect the unbroken $\mathrm{SU_L}(2) \times \mathrm{U(1)_Y}$ global symmetries of the background fields. The geoSMEFT approach was developed by first determining the BFM gauge fixing in the SMEFT in Ref. [9]. The BFM Ward identities for the SMEFT were reported in Ref. [10].

Remarkably, the BFM Ward identities are, for the most part,[2] the natural and direct gen-

---

[1]Generally the canonically normalised SMEFT parameters consistently defined on the curved background manifold of the Higgs are denoted in this work with a bar superscript, such as $M_Z \to \bar{M}_Z, s_\theta \to s_{\bar{\theta}}$ etc..

[2]An exception is the modification of the tadpole terms dependence in the SMEFT Ward identities, due to the need to carefully treat two derivative operators involving the Higgs field.

eralization of the SM BFM Ward identities; with the SM parameters generalized to the curved field space Lagrangian terms in the geoSMEFT [10]. This supports the notion that the use of the BFM in the SMEFT is of increased importance. When a field theory does not have a physical non-trivial background field configuration, the use of the BFM is largely a choice of convenience in a calculation. In the SMEFT the physics is different, as it is an EFT with a non-trivial background manifold, namely, the Higgs taking on its vacuum expectation value ($\bar{v}_T$). As such, a BFM based approach to the SMEFT naturally and efficiently organizes the physics that is present, at higher orders in the power counting expansions, and the loop expansion. Considering the complexity of the SMEFT, the cross checks afforded in this approach are quite valuable to validate results and avoid subtle theoretical inconsistencies. Although subtle, such inconsistencies can introduce violations of background field symmetries (i.e. make it impossible to consistently incorporate the IR effect of the field space geometries) and dramatically impact conclusions drawn from experimental constraints, which are $S$ matrix elements that depend on a consistent projection of the field space geometry. For a discussion on one such subtlety in Electroweak precision data, with significant consequences to the SMEFT global fit effort, see Ref. [14].

The BFM Ward identities constrain n-point functions and the SMEFT masses, gauge couplings and mixing angles. As the higher dimensional operators in the SMEFT also obey the $SU(3) \times SU_L(2) \times U(1)_Y$ symmetry of the SM, the one loop Ward identities formulated in the BFM are respected operator by operator in the SMEFT. In this paper, we demonstrate this is indeed the case. We explicitly verify a set of these identities (relating one and two point functions) to one loop order, and demonstrate the manner in which various contributions combine to satisfy the BFM Ward identities of the SMEFT operator by operator, in a consistent formulation of this theory to $\mathcal{O}(\bar{v}_T^2/\Lambda^2\, g_{SM}^n/16\pi^2)$.

## 2  SMEFT and geoSMEFT

The SMEFT Lagrangian is defined as

$$\mathcal{L}_{\text{SMEFT}} = \mathcal{L}_{\text{SM}} + \mathcal{L}^{(d)}, \qquad \mathcal{L}^{(d)} = \sum_i \frac{C_i^{(d)}}{\Lambda^{d-4}} \mathcal{Q}_i^{(d)} \quad \text{for } d > 4. \tag{1}$$

The SM Lagrangian and conventions are consistent with Ref. [3, 15, 16]. The operators $\mathcal{Q}_i^{(d)}$ are labelled with a mass dimension $d$ superscript and multiply unknown Wilson coefficients $C_i^{(d)}$. Conventionally we define $\tilde{C}_i^{(d)} \equiv C_i^{(d)} \bar{v}_T^{d-4}/\Lambda^{d-4}$. The parameter $\bar{v}_T \equiv \sqrt{\langle 2H^\dagger H \rangle}$ in the SMEFT is defined as the minimum of the potential, including corrections due to higher-dimensional operators. We use the Warsaw basis [2] for $\mathcal{L}^{(6)}$ and otherwise geoSMEFT [3] for operator conventions. GeoSMEFT organizes the theory in terms of field-space connections $G_i$ multiplying composite operator forms $f_i$, represented schematically by

$$\mathcal{L}_{\text{SMEFT}} = \sum_i G_i(I, A, \phi \dots) f_i, \tag{2}$$

where $G_i$ depend on the group indices $I, A$ of the (non-spacetime) symmetry groups, and the scalar field coordinates of the composite operators, except powers of $D^\mu H$, which are grouped into $f_i$. The field-space connections depend on the coordinates of the Higgs scalar doublet expressed in terms of real scalar field coordinates, $\phi_I = \{\phi_1, \phi_2, \phi_3, \phi_4\}$, with normalization

$$H(\phi_I) = \frac{1}{\sqrt{2}} \begin{bmatrix} \phi_2 + i\phi_1 \\ \phi_4 - i\phi_3 \end{bmatrix}. \tag{3}$$

The gauge boson field coordinates are defined as $\mathcal{W}^A = \{W^1, W^2, W^3, B\}$ with $A = \{1, 2, 3, 4\}$. The corresponding general coupling in the SM is $\alpha_A = \{g_2, g_2, g_2, g_1\}$. The mass eigenstate field coordinates are $\mathcal{A}^A = \{\mathcal{W}^+, \mathcal{W}^-, \mathcal{Z}, \mathcal{A}\}$.

The geometric Lagrangian parameters that appear in the Ward identities are functions of the field-space connections. Of particular importance are the field space connections $h_{IJ}, g_{AB}$ which we refer to as metrics in this work. These metrics are defined at all orders in the geoSMEFT organization of the SMEFT operator expansion as

$$h_{IJ}(\phi) = \frac{g_{\mu\nu}}{d} \left. \frac{\delta^2 \mathcal{L}_{\text{SMEFT}}}{\delta(D_\mu \phi)^I \, \delta(D_\nu \phi)^J} \right|_{\mathcal{L}(\alpha, \beta \cdots) \to 0}, \tag{4}$$

and

$$g_{AB}(\phi) = \frac{-2 \, g_{\mu\nu} g_{\sigma\rho}}{d^2} \left. \frac{\delta^2 \mathcal{L}_{\text{SMEFT}}}{\delta \mathcal{W}^A_{\mu\sigma} \, \delta \mathcal{W}^B_{\nu\rho}} \right|_{\mathcal{L}(\alpha, \beta \cdots) \to 0, \text{CP-even}}. \tag{5}$$

The notation $\mathcal{L}(\alpha, \beta \cdots)$ corresponds to non-trivial Lorentz-index-carrying Lagrangian terms and spin connections, e.g. $(D^\mu \Phi)^K$ and $W^A_{\mu\nu}$. The explicit form of the metrics are given in Ref. [3]. Here $d$ is the spacetime dimension. The matrix square roots of these field space connections are $\sqrt{g}_{AB} = \langle g_{AB} \rangle^{1/2}$, and $\sqrt{h}_{IJ} = \langle h_{IJ} \rangle^{1/2}$. The SMEFT perturbations are small corrections to the SM, so the field-space connections are positive semi-definite matrices, with unique positive semi-definite square roots.[3]

The transformation of the gauge fields, gauge parameters and scalar fields into mass eigenstates in the SMEFT is given *at all orders in the* $\bar{v}_T/\Lambda$ *expansion* by

$$\hat{\mathcal{W}}^{A,\nu} = \sqrt{g}^{AB} U_{BC} \hat{\mathcal{A}}^{C,\nu}, \tag{6}$$

$$\alpha^A = \sqrt{g}^{AB} U_{BC} \beta^C, \tag{7}$$

$$\hat{\phi}^J = \sqrt{h}^{JK} V_{KL} \hat{\Phi}^L, \tag{8}$$

with $\hat{\mathcal{A}}^C = (\hat{\mathcal{W}}^+, \hat{\mathcal{W}}^-, \hat{\mathcal{Z}}, \hat{\mathcal{A}})$, $\hat{\Phi}^L = \{\hat{\Phi}^+, \hat{\Phi}^-, \hat{\chi}, \hat{H}\}$. $\beta^C$ is obtained directly from $\alpha^A$ (defined above) and $U_{BC}$. The transformation of the quantum fields is of the same form. The matrices $U, V$ are unitary, and given by

$$U_{BC} = \begin{bmatrix} \frac{1}{\sqrt{2}} & \frac{1}{\sqrt{2}} & 0 & 0 \\ \frac{i}{\sqrt{2}} & \frac{-i}{\sqrt{2}} & 0 & 0 \\ 0 & 0 & c_{\bar{\theta}} & s_{\bar{\theta}} \\ 0 & 0 & -s_{\bar{\theta}} & c_{\bar{\theta}} \end{bmatrix}, \qquad V_{JK} = \begin{bmatrix} \frac{-i}{\sqrt{2}} & \frac{i}{\sqrt{2}} & 0 & 0 \\ \frac{1}{\sqrt{2}} & \frac{1}{\sqrt{2}} & 0 & 0 \\ 0 & 0 & -1 & 0 \\ 0 & 0 & 0 & 1 \end{bmatrix}.$$

These matricies $U, V$ are rotations; i.e. orthogonal matricies whose transpose is equal to the matrix inverse. The short hand combinations

$$\mathcal{U}^A_C = \sqrt{g}^{AB} U_{BC}, \qquad\qquad (\mathcal{U}^{-1})^D_F = U^{DE} \sqrt{g}_{EF},$$

$$\mathcal{V}^A_C = \sqrt{h}^{AB} V_{BC}, \qquad\qquad (\mathcal{V}^{-1})^D_F = V^{DE} \sqrt{h}_{EF},$$

are useful to define as they perform the mass eigenstate rotation for the vector and scalar fields, and bring the corresponding kinetic term to canonical form, including higher-dimensional-operator corrections. As can be directly verified, the combined operation is not an orthogonal matrix whose transpose is equal to the matrix inverse; i.e. $\mathcal{U}^A_C, \mathcal{V}^A_C$ are not rotations. Although the transformation between mass and canonically normalized weak eigenstates are properly and formally rotations in the SM, this is no longer the case in the SMEFT.

---

[3]Note that $\sqrt{g}^{AB} \sqrt{g}_{BC} \equiv \delta^A_C$ and $\sqrt{h}^{IJ} \sqrt{h}_{JK} \equiv \delta^I_K$.

# 3   Background Field Method, Gauge fixing and Ward identities

The BFM [17–19] is a theoretical approach to gauge fixing a quantum field theory in a manner that leaves the effective action invariant under background field gauge transformations. To this end, the fields are split into quantum (un-hated) and classical (hatted) background fields: $F \to F + \hat{F}$. The classical fields are associated with the external states of the $S$-matrix in an LSZ procedure [20], and a gauge fixing term is defined so that the effective action is unchanged under a local gauge transformation of the background fields in conjunction with a linear change of variables on the quantum fields, see Ref. [19].

In the BFM, relationships between Lagrangian parameters due to unbroken background $SU_L(2) \times U(1)_Y$ symmetry then follow a "naive" (classical) expectation when quantizing the theory. These are the BFM Ward identities. In the case of the SMEFT, the naive BFM Ward identities of the SM are upgraded to involve the canonically normalized Lagrangian parameters (i.e. barred parameters) defined in the geoSMEFT by using the field space connections.

The BFM generating functional of the SMEFT is given by

$$Z[\hat{F}, J] = \int \mathcal{D}F \det\left[\frac{\Delta \mathcal{G}^A}{\Delta \alpha^B}\right] e^{i \int dx^4 \left(S[F+\hat{F}] + \mathcal{L}_{\text{GF}} + \text{source terms}\right)}.$$

The generating functional is integrated over the quantum field configurations via $\mathcal{D}F$, with $F$ field coordinates describing all long-distance propagating states. The sources $J$ only couple to the quantum fields [21]. The issue of gauge fixing the SMEFT in the BFM was discussed as a novel challenge in Ref. [22] (see also Refs. [23–25]). The core issue to utilizing the BFM in the SMEFT (to calculate complete dependence on IR quantities such as masses) is to define a gauge fixing procedure in the presence of higher dimensional operators, while preserving background field gauge invariance. Ref. [9] reported that such a gauge fixing term is uniquely

$$\mathcal{L}_{\text{GF}} = -\frac{\hat{g}_{AB}}{2\xi} \mathcal{G}^A \mathcal{G}^B, \qquad \mathcal{G}^X \equiv \partial_\mu \mathcal{W}^{X,\mu} - \tilde{\epsilon}^X_{CD} \hat{\mathcal{W}}^C_\mu \mathcal{W}^{D,\mu} + \frac{\xi}{2} \hat{g}^{XC} \phi^I \hat{h}_{IK} \tilde{\gamma}^K_{C,J} \hat{\phi}^J. \qquad (9)$$

Here $\hat{g}$ and $\hat{h}$ are the background field values of the metrics, as indicated with the hat superscript. See Ref. [9] for more details. This approach to gauge fixing has an intuitive interpretation. The fields are gauge fixed on the curved Higgs field space defined by the SMEFT (field) power counting expansion (i.e. in $\bar{v}_T/\Lambda$). This is done by upgrading the naive squares of fields in the gauge fixing term, to less-naive contractions of fields through the Higgs field space metrics $g_{AB}, h_{IK}$. Such contractions characterize the curved Higgs field space geometry the theory is being quantized on to define the correlation functions. When the field space metrics are trivialized to their values in the SM: $\hat{h}_{IJ} = \delta_{IJ}$ and $\hat{g}_{AB} = \delta_{AB}$. The field space manifold is no longer curved due to SMEFT corrections in this $\bar{v}_T/\Lambda \to 0$ limit. The gauge fixing term in the Background Field Method then simplifies to that in the SM, as given in Ref. [26–28].

The Faddeev-Popov ghost term, derived from Eqn. 9 is [9]

$$\mathcal{L}_{\text{FP}} = -\hat{g}_{AB} \bar{u}^B \left[ -\partial^2 \delta^A_C - \overleftarrow{\partial}_\mu \tilde{\epsilon}^A_{DC} (\mathcal{W}^{D,\mu} + \hat{\mathcal{W}}^{D,\mu}) + \tilde{\epsilon}^A_{DC} \hat{\mathcal{W}}^D_\mu \overrightarrow{\partial}^\mu \right. \qquad (10)$$

$$\left. -\tilde{\epsilon}^A_{DE} \tilde{\epsilon}^E_{FC} \hat{\mathcal{W}}^D_\mu (\mathcal{W}^{F,\mu} + \hat{\mathcal{W}}^{F,\mu}) - \frac{\xi}{4} \hat{g}^{AD} (\phi^J + \hat{\phi}^J) \tilde{\gamma}^I_{C,J} \hat{h}_{IK} \tilde{\gamma}^K_{D,L} \hat{\phi}^L \right] u^C.$$

Our notation is such that the covariant derivative acting on the bosonic fields of the SM in the doublet and real representations respectively is [9]

$$D^\mu H = (\partial^\mu + i g_2 W^{a,\mu} \sigma_a/2 + i g_1 y_h B^\mu) H, \qquad (11)$$

$$(D^\mu \phi)^I = (\partial^\mu \delta^I_J - \frac{1}{2} \mathcal{W}^{A,\mu} \tilde{\gamma}^I_{A,J}) \phi^J, \qquad (12)$$

with symmetry generators for the real scalar manifold $\tilde{\gamma}^I_{A,j}$ (see Ref. [3,9] for the explicit forms of the generators). Here $\sigma_a$ are the Pauli matrices and $a = \{1, 2, 3\}$, $y_h$ is the Hypercharge of the Higgs field. The structure constants (that absorb gauge coupling parameters) are

$$
\begin{aligned}
\tilde{\epsilon}^A_{BC} &= g_2 \, \epsilon^A_{BC}, \quad \text{with } \tilde{\epsilon}^1_{23} = g_2, \\
\tilde{\gamma}^I_{A,J} &= \begin{cases} g_2 \, \gamma^I_{A,J}, & \text{for } A = 1, 2, 3 \\ g_1 \gamma^I_{A,J}, & \text{for } A = 4. \end{cases}
\end{aligned}
\tag{13}
$$

For infinitesimal local gauge parameters $\delta\hat{\alpha}_A(x)$ the BF gauge transformations are

$$
\begin{aligned}
\delta\,\hat{\phi}^I &= -\delta\hat{\alpha}^A \frac{\tilde{\gamma}^I_{A,J}}{2} \hat{\phi}^J, \\
\delta\hat{\mathcal{W}}^{A,\mu} &= -(\partial^\mu \delta^A_B + \tilde{\epsilon}^A_{BC} \, \hat{\mathcal{W}}^{C,\mu}) \delta\hat{\alpha}^B, \\
\delta\hat{h}_{IJ} &= \hat{h}_{KJ} \frac{\delta\hat{\alpha}^A \tilde{\gamma}^K_{A,I}}{2} + \hat{h}_{IK} \frac{\delta\hat{\alpha}^A \tilde{\gamma}^K_{A,J}}{2}, \\
\delta\hat{g}_{AB} &= \hat{g}_{CB} \, \tilde{\epsilon}^C_{DA} \, \delta\hat{\alpha}^D + \hat{g}_{AC} \, \tilde{\epsilon}^C_{DB} \, \delta\hat{\alpha}^D, \\
\delta\mathcal{G}^X &= -\tilde{\epsilon}^X_{AB} \, \delta\hat{\alpha}^A \mathcal{G}^B, \\
\delta f_i &= \Lambda^j_{A,i} \, \hat{\alpha}^A f_j, \\
\delta\bar{f}_i &= \hat{\alpha}^A \bar{f}_j \bar{\Lambda}^j_{A,i}.
\end{aligned}
\tag{14}
$$

The BFM Ward identities follow from the invariance of $\Gamma[\hat{F}, 0]$ under background-field gauge transformations,

$$
\frac{\delta\Gamma[\hat{F}, 0]}{\delta\hat{\alpha}^B} = 0.
\tag{15}
$$

In position space, the identities are [9]

$$
0 = \left(\partial^\mu \delta^A_B - \tilde{\epsilon}^A_{BC} \, \hat{\mathcal{W}}^{C,\mu}\right) \frac{\delta\Gamma}{\delta\hat{\mathcal{W}}^\mu_A} - \frac{\tilde{\gamma}^I_{B,J}}{2} \hat{\phi}^J \frac{\delta\Gamma}{\delta\hat{\phi}^I} + \sum_j \left(\bar{f}_j \bar{\Lambda}^j_{B,i} \frac{\delta\Gamma}{\delta\bar{f}_i} - \frac{\delta\Gamma}{\delta f_i} \Lambda^i_{B,j} f_j\right).
\tag{16}
$$

The structure constants and generators, transformed to those corresponding to the mass eigenstates, are defined using bold text as

$$
\epsilon^C_{GY} = (\mathcal{U}^{-1})^C_A \tilde{\epsilon}^A_{DE} \mathcal{U}^D_G \mathcal{U}^E_Y, \qquad\qquad \boldsymbol{\gamma}^I_{G,L} = \frac{1}{2} \tilde{\gamma}^I_{A,L} \mathcal{U}^A_G
$$
$$
\boldsymbol{\Lambda}^i_{X,j} = \Lambda^i_{A,j} \mathcal{U}^A_X.
$$

The background-field gauge transformations in the mass eigenstates are

$$
\begin{aligned}
\delta\hat{\mathcal{A}}^{C,\mu} &= -\left[\partial^\mu \delta^C_G + \epsilon^C_{GY} \hat{\mathcal{A}}^{Y,\mu}\right] \delta\hat{\beta}^G \\
\delta\hat{\Phi}^K &= -(\mathcal{V}^{-1})^K_I \boldsymbol{\gamma}^I_{G,L} \mathcal{V}^L_N \hat{\Phi}^N \delta\hat{\beta}^G.
\end{aligned}
\tag{17}
$$

The Ward identities are then expressed compactly as [9]

$$
\begin{aligned}
0 &= \frac{\delta\Gamma}{\delta\hat{\beta}^G} \\
&= \partial^\mu \frac{\delta\Gamma}{\delta\hat{\mathcal{A}}^{X,\mu}} + \sum_j \left(\bar{f}_j \bar{\boldsymbol{\Lambda}}^j_{X,i} \frac{\delta\Gamma}{\delta\bar{f}_i} - \frac{\delta\Gamma}{\delta f_i} \boldsymbol{\Lambda}^i_{X,j} f_j\right) - \frac{\delta\Gamma}{\delta\hat{\mathcal{A}}^{C\mu}} \epsilon^C_{XY} \hat{\mathcal{A}}^{Y\mu} - \frac{\delta\Gamma}{\delta\hat{\Phi}^K} (\mathcal{V}^{-1})^K_I \boldsymbol{\gamma}^I_{X,L} \mathcal{V}^L_N \hat{\Phi}^N.
\end{aligned}
\tag{18}
$$

In this manner, the "naive" form of the Ward identities is maintained. The descending relationships between n-point functions encode the constraints of the unbroken (but non-manifest in the mass eigenstates) $SU(2)_L \times U(1)_Y$ symmetry that each operator in the SMEFT respects.

# 4 Background Field Ward Identities

The results of this work are the SMEFT extension of the treatment of the Electroweak Standard Model in the BFM, as developed in Refs. [26–33]. Our results (with appropriate notational redefinitions) simplify to those reported in these past works in the limit $\bar{v}_T/\Lambda \to 0$. The background Higgs field $\hat{H}$ takes on this vacuum expectation value, while the quantum Higgs field has vanishing expectation value

$$\hat{H}(\phi_I) = \frac{1}{\sqrt{2}}\begin{bmatrix} \hat{\phi}_2 + i\hat{\phi}_1 \\ \bar{v}_T + \hat{\phi}_4 - i\hat{\phi}_3 \end{bmatrix}, \qquad\qquad H(\phi_I) = \frac{1}{\sqrt{2}}\begin{bmatrix} \phi_2 + i\phi_1 \\ \phi_4 - i\phi_3 \end{bmatrix}.$$

In the remainder of this paper we verify that a set of the Ward identities hold at one loop order. This requires some notation. Our convention is that all momentum are incoming (here denoted with $k^\mu$) and we define short hand notation as in Refs. [28–33]

$$-i\Gamma_{\mu\nu}^{\hat{V},\hat{V}'}(k) = \left(-g_{\mu\nu}k^2 + k_\mu k_\nu + g_{\mu\nu}\bar{M}_{\hat{V}}^2\right)\delta^{\hat{V}\hat{V}'} + \left(-g_{\mu\nu} + \frac{k_\mu k_\nu}{k^2}\right)\Sigma_T^{\hat{V},\hat{V}'} - \frac{k_\mu k_\nu}{k^2}\Sigma_L^{\hat{V},\hat{V}'}, \quad (19)$$

$$\frac{\delta^2\Gamma}{\delta\hat{\mathcal{A}}^{4\mu}\delta\hat{\phi}^3} = ik^\mu\Sigma^{\hat{\mathcal{A}}\hat{\chi}}(k^2), \tag{20}$$

$$\frac{\delta^2\Gamma}{\delta\hat{\mathcal{A}}^{4\mu}\delta\hat{\phi}^4} = ik^\mu\Sigma^{\hat{\mathcal{A}}\hat{\mathcal{H}}}(k^2), \tag{21}$$

$$\frac{\delta^2\Gamma}{\delta\hat{\phi}^3\delta\hat{\mathcal{A}}^{3\nu}} = -\frac{\delta^2\Gamma}{\delta\hat{\mathcal{A}}^{3\nu}\delta\hat{\phi}^3} = ik^\nu\left[i\bar{M}_{\mathcal{Z}} + \Sigma^{\hat{Z}\hat{\chi}}(k^2)\right], \tag{22}$$

$$\frac{\delta^2\Gamma}{\delta\hat{\phi}^3\delta\hat{\phi}^3} = ik^2 + i\Sigma^{\hat{\chi}\hat{\chi}}(k^2), \tag{23}$$

$$\frac{\delta^2\Gamma}{\delta\hat{\phi}^\pm\delta\hat{\mathcal{W}}^{\mp\nu}} = -ik^\nu\left[\pm\bar{M}_W + \Sigma^{\hat{\phi}^\pm\hat{\mathcal{W}}^\mp}(k^2)\right], \tag{24}$$

$$\frac{\delta^2\Gamma}{\delta\hat{\mathcal{W}}^{\pm\nu}\delta\hat{\phi}^\mp} = ik^\nu\left[\mp\bar{M}_W + \Sigma^{\hat{\mathcal{W}}^\pm\hat{\phi}^\mp}(k^2)\right], \tag{25}$$

$$\frac{\delta^2\Gamma}{\delta\hat{\phi}^+\delta\hat{\phi}^-} = ik^2 + i\Sigma^{\hat{\phi}^+\hat{\phi}^-}(k^2), \tag{26}$$

$$\frac{\delta^2\Gamma}{\delta\hat{H}\delta\hat{H}} = i(k^2 - \bar{m}_H^2) + i\Sigma^{\hat{H}\hat{H}}(k^2). \tag{27}$$

The two point function mass eigenstate SMEFT Ward identities in the BFM are [10]

$$0 = \partial^\mu\frac{\delta^2\Gamma}{\delta\hat{\mathcal{A}}^{4\mu}\delta\hat{\mathcal{A}}^{Y\nu}}, \tag{28}$$

$$0 = \partial^\mu\frac{\delta^2\Gamma}{\delta\hat{\mathcal{A}}^{4\mu}\delta\hat{\phi}^I}, \tag{29}$$

$$0 = \partial^\mu\frac{\delta^2\Gamma}{\delta\hat{\mathcal{A}}^{3\mu}\delta\hat{\mathcal{A}}^{Y\nu}} - \bar{M}_{\mathcal{Z}}\frac{\delta^2\Gamma}{\delta\hat{\phi}^3\delta\hat{\mathcal{A}}^{Y\nu}}, \tag{30}$$

and

$$0 = \partial^\mu \frac{\delta^2 \Gamma}{\delta \hat{\mathcal{A}}^{3\mu} \delta \hat{\Phi}^I} - \bar{M}_{\mathcal{Z}} \frac{\delta^2 \Gamma}{\delta \hat{\Phi}^3 \delta \hat{\Phi}^I} \tag{31}$$
$$+ \frac{\bar{g}_Z}{2} \frac{\delta \Gamma}{\delta \hat{\Phi}^4} \left[ \left( \sqrt{h}_{[4,4]} \sqrt{h}^{[3,3]} - \sqrt{h}_{[4,3]} \sqrt{h}^{[4,3]} \right) \delta_I^3 - \left( \sqrt{h}_{[4,4]} \sqrt{h}^{[3,4]} - \sqrt{h}_{[4,3]} \sqrt{h}^{[4,4]} \right) \delta_I^4 \right],$$

$$0 = \partial^\mu \frac{\delta^2 \Gamma}{\delta \hat{\mathcal{W}}^{\pm\mu} \delta \hat{\mathcal{A}}^{Y\nu}} \pm i \bar{M}_W \frac{\delta^2 \Gamma}{\delta \hat{\Phi}^\pm \delta \hat{\mathcal{A}}^{Y\nu}}, \tag{32}$$

$$0 = \partial^\mu \frac{\delta^2 \Gamma}{\delta \hat{\mathcal{W}}^{\pm\mu} \delta \hat{\Phi}^I} \pm i \bar{M}_W \frac{\delta^2 \Gamma}{\delta \hat{\Phi}^\pm \delta \hat{\Phi}^I} \mp \frac{i \bar{g}_2}{4} \frac{\delta \Gamma}{\delta \hat{\Phi}^4} \left( \sqrt{h}_{[4,4]} \mp i \sqrt{h}_{[4,3]} \right) \times \tag{33}$$
$$\left[ (\sqrt{h}^{[1,1]} + \sqrt{h}^{[2,2]} \mp i \sqrt{h}^{[1,2]} \pm i \sqrt{h}^{[2,1]}) \delta_I^\mp \mp (\sqrt{h}^{[1,1]} - \sqrt{h}^{[2,2]} \pm i \sqrt{h}^{[1,2]} \pm i \sqrt{h}^{[2,1]}) \delta_I^\pm \right].$$

To utilize these definitions, note that sign dependence of $k^\mu$ being always incoming in the case of charged fields leads to several implicit sign conventions that must be respected to establish the Ward identities. From these identities, it follows that

$$\Sigma_L^{\hat{\mathcal{A}}\hat{\mathcal{A}}}(k^2) = 0, \qquad\qquad \Sigma_T^{\hat{\mathcal{A}}\hat{\mathcal{A}}}(0) = 0, \tag{34}$$
$$\Sigma_L^{\hat{\mathcal{A}}\hat{\mathcal{Z}}}(k^2) = 0, \qquad\qquad \Sigma_T^{\hat{\mathcal{A}}\hat{\mathcal{Z}}}(0) = 0, \tag{35}$$

and

$$\Sigma^{\hat{\mathcal{A}}\hat{\chi}}(k^2) = 0, \qquad\qquad \Sigma^{\hat{\mathcal{A}}\hat{\mathcal{H}}}(k^2) = 0. \tag{36}$$

Limiting the evaluation of the field space metrics to $\mathcal{L}^{(6)}$ corrections in the Warsaw basis [2], further identities that directly follow are

$$0 = \Sigma_L^{\hat{\mathcal{Z}}\hat{\mathcal{Z}}}(k^2) - i \bar{M}_{\mathcal{Z}} \Sigma^{\hat{\mathcal{Z}}\hat{\chi}}(k^2), \tag{37}$$
$$0 = k^2 \Sigma^{\hat{\mathcal{Z}}\hat{\chi}}(k^2) - i \bar{M}_{\mathcal{Z}} \Sigma^{\hat{\chi}\hat{\chi}}(k^2) + i \frac{\bar{g}_Z}{2} T^H \left( 1 - \tilde{C}_{H\square} \right), \tag{38}$$

and

$$0 = \Sigma_L^{\hat{\mathcal{W}}^\pm \hat{\mathcal{W}}^\mp}(k^2) \pm \bar{M}_W \Sigma^{\hat{\Phi}^\pm \hat{\mathcal{W}}^\mp}(k^2), \tag{39}$$
$$0 = k^2 \Sigma^{\hat{\mathcal{W}}^\pm \hat{\Phi}^\mp}(k^2) \pm \bar{M}_W \Sigma^{\hat{\Phi}^\pm \hat{\Phi}^\mp}(k^2) \mp \frac{\bar{g}_2}{2} T^H \left( 1 - \tilde{C}_{H\square} + \frac{\tilde{C}_{HD}}{4} \right). \tag{40}$$

Note the appearance of the two derivative operators involving the Higgs field modifying the tadpole terms $T^H = -i \delta \Gamma / \delta \hat{H}$ fixing the vev. It is important to include such corrections, which are a consistency condition due to the background field geometry the SMEFT is quantized on.

Several of the remaining two point functions vanish exactly, and the corresponding Ward identities are trivially satisfied. The geometric SMEFT Lagrangian parameters to $\mathcal{L}^{(6)}$ appearing in the Ward identities are the geometric SMEFT masses [15]

$$\bar{M}_W^2 = \frac{\bar{g}_2^2 \bar{v}_T^2}{4}, \tag{41}$$

$$\bar{M}_{\mathcal{Z}}^2 = \frac{\bar{v}_T^2}{4}(\bar{g}_1^2 + \bar{g}_2^2) + \frac{1}{8} \bar{v}_T^2 (\bar{g}_1^2 + \bar{g}_2^2) \tilde{C}_{HD} + \frac{1}{2} \bar{v}_T^2 \bar{g}_1 \bar{g}_2 \tilde{C}_{HWB}, \tag{42}$$

$$\bar{m}_h^2 = 2\lambda \bar{v}_T^2 \left[ 1 - 3 \frac{\tilde{C}_H}{2\lambda} + 2 \left( \tilde{C}_{H\square} - \frac{\tilde{C}_{HD}}{4} \right) \right], \tag{43}$$

and the geometric SMEFT couplings

$$\bar{e} = \frac{\overline{g}_1 \overline{g}_2}{\sqrt{\overline{g}_1^2 + \overline{g}_2^2}} \left[ 1 - \frac{\overline{g}_1 \overline{g}_2}{\overline{g}_1^2 + \overline{g}_2^2} \tilde{C}_{HWB} \right], \qquad \bar{g}_Z = \sqrt{\overline{g}_1^2 + \overline{g}_2^2} + \frac{\overline{g}_1 \overline{g}_2}{\sqrt{\overline{g}_1^2 + \overline{g}_2^2}} \tilde{C}_{HWB}, \quad (44)$$

$$\overline{g}_1 = g_1(1 + \tilde{C}_{HB}), \qquad\qquad\qquad \overline{g}_2 = g_2(1 + \tilde{C}_{HW}). \qquad (45)$$

These parameters are defined at all orders in the $\bar{v}_T/\Lambda$ expansion in Ref. [3,11], and we stress the Ward identities hold at all orders in the $\bar{v}_T/\Lambda$ expansion, and also hold for cross terms in the perturbative expansion and $\bar{v}_T/\Lambda$ expansion. As such, the Ward identities provide a powerful and important cross check of non-perturbative and perturbative results in the SMEFT.

## 4.1 SM results; Bosonic loops

We verify the Ward identities at the level of divergent one-loop contributions to the various $n$-point functions. In the case of the SM, we confirm the results of Refs. [28–33] and reiterate these results here for a common notation and due to their contributions to the SMEFT Ward identities. We focus on two point functions involving the gauge fields due to the role that the scalar and gauge boson field space metrics have as the field space geometry modifies the Ward identities into those of the SMEFT. The results (using $d = 4 - 2\epsilon$ in dim. reg.) are

$$\left[ \Sigma_T^{\hat{A}\hat{A}}(k^2) \right]_{SM}^{div} = \frac{g_1^2 g_2^2}{(g_1^2 + g_2^2)} k^2 \left( \frac{-7}{16\pi^2\epsilon} \right), \qquad (46)$$

$$\left[ \Sigma_L^{\hat{A}\hat{A}}(k^2) \right]_{SM}^{div} = 0, \qquad (47)$$

$$\left[ \Sigma_T^{\hat{A}\hat{Z}}(k^2) \right]_{SM}^{div} = -\frac{g_1 g_2}{(g_1^2 + g_2^2)} k^2 \left( \frac{43g_2^2 + g_1^2}{96\pi^2\epsilon} \right), \qquad (48)$$

$$\left[ \Sigma_L^{\hat{A}\hat{Z}}(k^2) \right]_{SM}^{div} = 0, \qquad (49)$$

and

$$\left[ \Sigma_T^{\hat{Z}\hat{Z}}(k^2) \right]_{SM}^{div} = \frac{8k^2(g_1^4 - 43g_2^4) + 3(\xi + 3)\bar{v}_T^2(g_1^2 + g_2^2)^2(g_1^2 + 3g_2^2)}{768\,\pi^2\,\epsilon\,(g_1^2 + g_2^2)}, \qquad (50)$$

$$\left[ \Sigma_L^{\hat{Z}\hat{Z}}(k^2) \right]_{SM}^{div} = \frac{(\xi + 3)\bar{v}_T^2(g_1^2 + g_2^2)(g_1^2 + 3g_2^2)}{256\pi^2\epsilon}, \qquad (51)$$

$$\left[ \Sigma^{\hat{Z}\hat{\chi}}(k^2) \right]_{SM}^{div} = -i\frac{(\xi + 3)\bar{v}_T \sqrt{g_1^2 + g_2^2}(g_1^2 + 3g_2^2)}{128\pi^2\epsilon}, \qquad (52)$$

$$\left[ \Sigma_T^{\hat{W}^\pm \hat{W}^\mp}(k^2) \right]_{SM}^{div} = \frac{g_2^2(3(\xi + 3)\bar{v}_T^2(g_1^2 + 3g_2^2) - 344k^2)}{768\pi^2\epsilon}, \qquad (53)$$

$$\left[ \Sigma_L^{\hat{W}^\pm \hat{W}^\mp}(k^2) \right]_{SM}^{div} = \frac{g_2^2(\xi + 3)\bar{v}_T^2(g_1^2 + 3g_2^2)}{256\pi^2\epsilon}, \qquad (54)$$

$$\left[ \Sigma^{\hat{\phi}^+ \hat{W}^-}(k^2) \right]_{SM}^{div} = -\left[ \Sigma^{\hat{\phi}^- \hat{W}^+}(k^2) \right]_{SM}^{div} = -\frac{g_2(\xi + 3)\bar{v}_T(g_1^2 + 3g_2^2)}{128\pi^2\epsilon}, \qquad (55)$$

$$\left[ \Sigma^{\hat{\chi}\hat{\chi}}(k^2) \right]_{SM}^{div} = \left[ \Sigma^{\hat{\phi}^+ \hat{\phi}^-}(k^2) \right]_{SM}^{div} = \frac{1}{\bar{v}_T}\left[ T^H \right]_{SM}^{div} - \frac{(\xi + 3)k^2(g_1^2 + 3g_2^2)}{64\pi^2\epsilon}, \qquad (56)$$

$$\left[ T^H \right]_{SM}^{div} = \frac{\bar{v}_T^3(3g_1^4 + 9g_2^4 + 96\lambda^2 + 12g_2^2\lambda\xi + g_1^2(6g_2^2 + 4\lambda\xi))}{256\,\pi^2\,\epsilon}. \qquad (57)$$

Reducing to the SM limit the SMEFT Ward ID ($\Lambda \to \infty, \bar{v}_T \to v$) yields the corresponding SM Ward ID, consistent with Refs. [28–33]. These expressions satisfy the SM Ward identities.

The fermion self energies in the SM, and the fermionic contributions to the bosonic two point functions are suppressed here.

## 4.2 SM results; Fermion loops

Unlike the contributions to the bosonic one and two point functions discussed in the previous section, the contributions from fermion loops depend on the number of fermion generations. We discuss these contributions in a factorized fashion in the SM and the SMEFT for this reason. The bosonic one and two point functions contributions in the SM from fermion loops are shown in Fig. 1, which give the results

$$\left[\Sigma_T^{\hat{\mathcal{A}}\hat{\mathcal{A}}}(k^2)\right]_{SM}^{div} = \frac{g_1^2 g_2^2}{(g_1^2 + g_2^2)}\left(\frac{32}{9}\right)\frac{k^2}{16\pi^2\epsilon}\,n, \tag{58}$$

$$\left[\Sigma_L^{\hat{\mathcal{A}}\hat{\mathcal{A}}}(k^2)\right]_{SM}^{div} = 0, \tag{59}$$

$$\left[\Sigma_T^{\hat{\mathcal{A}}\hat{\mathcal{Z}}}(k^2)\right]_{SM}^{div} = -\frac{20\,g_1^3\,g_2 - 12\,g_1\,g_2^3}{9(g1^2 + g2^2)}\frac{k^2}{16\pi^2\epsilon}\,n, \tag{60}$$

$$\left[\Sigma_L^{\hat{\mathcal{A}}\hat{\mathcal{Z}}}(k^2)\right]_{SM}^{div} = 0, \tag{61}$$

$$\left[\Sigma_T^{\hat{\mathcal{W}}^+\hat{\mathcal{W}}^-}(k^2)\right]_{SM}^{div} = \frac{4\,g_2^2}{3}\frac{k^2}{16\,\pi^2\,\epsilon}\,n - \sum_\psi\frac{N_C^\psi\,m_\psi^2\,g_2^2}{32\pi^2\epsilon}, \tag{62}$$

$$\left[\Sigma_L^{\hat{\mathcal{W}}^+\hat{\mathcal{W}}^-}(k^2)\right]_{SM}^{div} = -\sum_\psi\frac{N_C^\psi\,m_\psi^2\,g_2^2}{32\pi^2\epsilon}, \tag{63}$$

$$\left[\Sigma_T^{\hat{\mathcal{Z}}\hat{\mathcal{Z}}}(k^2)\right]_{SM}^{div} = \frac{5g_1^4 + 3g_2^4}{g_1^2 + g_2^2}\frac{k^2}{36\pi^2\epsilon}\,n - \sum_\psi N_C^\psi\frac{m_\psi^2\,(g_1^2 + g_2^2)}{32\pi^2\epsilon}, \tag{64}$$

$$\left[\Sigma_L^{\hat{\mathcal{Z}}\hat{\mathcal{Z}}}(k^2)\right]_{SM}^{div} = -\sum_\psi\frac{N_C^\psi\,m_\psi^2\,(g_1^2 + g_2^2)}{32\pi^2\epsilon}, \tag{65}$$

$$\left[\Sigma^{\hat{\mathcal{Z}}\hat{\chi}}(k^2)\right]_{SM}^{div} = -i\sum_\psi N_C^\psi\,Y_\psi^2\,\bar{v}_T\,\frac{\sqrt{g_1^2 + g_2^2}}{32\,\pi^2\,\epsilon}, \tag{66}$$

$$\left[\Sigma^{\hat{\phi}^+\hat{\mathcal{W}}^-}(k^2)\right]_{SM}^{div} = -\left[\Sigma^{\hat{\phi}^-\hat{\mathcal{W}}^+}(k^2)\right]_{SM}^{div} = \sum_\psi N_C^\psi\,Y_\psi^2\,\bar{v}_T\,\frac{g_2}{32\,\pi^2\,\epsilon}, \tag{67}$$

$$\left[\Sigma^{\hat{\chi}\hat{\chi}}(k^2)\right]_{SM}^{div} = \frac{k^2}{16\,\pi^2\,\epsilon}\sum_\psi N_C^\psi\,Y_\psi^2 - \frac{\bar{v}_T^2}{16\,\pi^2\,\epsilon}\sum_\psi N_C^\psi\,Y_\psi^4, \tag{68}$$

$$\left[\Sigma^{\hat{\phi}^+\hat{\phi}^-}(k^2)\right]_{SM}^{div} = \frac{k^2}{16\,\pi^2\,\epsilon}\sum_\psi N_C^\psi\,Y_\psi^2 - \frac{\bar{v}_T^2}{16\,\pi^2\,\epsilon}N_C^\psi\,Y_\psi^4, \tag{69}$$

$$\left[T^H\right]_{SM}^{div} = -\frac{\bar{v}_T^3}{16\,\pi^2\,\epsilon}\sum_\psi N_C^\psi\,Y_\psi^4. \tag{70}$$

Here $Y_\psi$ is the fermion $\psi$ Yukawa coupling, and $N_C^\psi = (3, 3, 1)$ for up quarks, down quarks and leptons respectively. $n$ sums over the generations and colours in each generation. These expressions, consistent with those in Ref. [28–33] satisfy the SM limit BFM Ward identities.

## 4.3 SMEFT results; Bosonic loops

Directly evaluating the diagrams in Fig. 1, with a full set of all possible higher dimensional operator insertions, we find the following for the SMEFT. The results have been determined automatically using a new code package for BFM based SMEFT calculations to one loop order. This code package is reported on in a companion paper [34]. The results have also been directly calculated by hand independently in a cross check and verification of the automated generation of results. In many cases, consistently modifying the SM parameters into those of the geoSMEFT leads to some intricate cancelations in Wilson coefficient dependence in a Feynman diagram, through modified Feynman rules in the BFM, and subsequently in the summation of the diagrams into the two point functions. Further cancelations, and non-trivial combinations of Wilson coefficient dependence, occurs combining the full two point functions, with the geoSMEFT lagrangian parameters that feed into the Ward identities. Such intricate cancelations follow from the unbroken background field symmetries.

### 4.3.1 Operator $Q_{HB}$

Defining the combinations of coupling which occur frequently for this operator as

$$\mathcal{P}^1_{CHB} = \frac{((g_1^4 + 3\,g_2^4)\xi + 4\,g_1^2\,g_2^2(\xi - 7) + 8\,(g_1^2 + g_2^2)\lambda)}{32\pi^2\epsilon}, \tag{71}$$

$$\mathcal{P}^2_{CHB} = \frac{(3 + \xi)(g_1^2 + 2g_2^2)}{32\pi^2\epsilon}, \tag{72}$$

$$\mathcal{P}^3_{CHB} = \frac{7(g_1^2 + 7g_2^2)}{48\pi^2\epsilon}, \tag{73}$$

$$\mathcal{P}^4_{CHB} = \frac{9(g_1^2 + g_2^2) + 4\lambda\,\xi}{128\pi^2\epsilon}, \tag{74}$$

the two point functions in the SMEFT are

$$\left[\Sigma_T^{\hat{\mathcal{A}}\hat{\mathcal{A}}}(k^2)\right]^{div}_{\tilde{C}_{HB}} = \tilde{C}_{HB}\,k^2\,\frac{g_2^2\,\mathcal{P}^1_{CHB}}{(g_1^2 + g_2^2)^2}, \tag{75}$$

$$\left[\Sigma_L^{\hat{\mathcal{A}}\hat{\mathcal{A}}}(k^2)\right]^{div}_{\tilde{C}_{HB}} = 0, \tag{76}$$

$$\left[\Sigma_T^{\hat{\mathcal{A}}\hat{\mathcal{Z}}}(k^2)\right]^{div}_{\tilde{C}_{HB}} = -\tilde{C}_{HB}\,k^2\left[\frac{g_1\,g_2\,\mathcal{P}^1_{CHB}}{(g_1^2 + g_2^2)^2} + \frac{g_1\,g_2\,\mathcal{P}^3_{CHB}}{2\,(g_1^2 + g_2^2)}\right], \tag{77}$$

$$\left[\Sigma_L^{\hat{\mathcal{A}}\hat{\mathcal{Z}}}(k^2)\right]^{div}_{\tilde{C}_{HB}} = 0, \tag{78}$$

$$\left[\Sigma_T^{\hat{\mathcal{Z}}\hat{\mathcal{Z}}}(k^2)\right]^{div}_{\tilde{C}_{HB}} = \tilde{C}_{HB}\left[k^2\,\frac{g_1^2\,\mathcal{P}^1_{CHB}}{(g_1^2 + g_2^2)^2} + k^2\,\frac{g_1^2\,\mathcal{P}^3_{CHB}}{(g_1^2 + g_2^2)} + \bar{v}_T^2\,\frac{g_1^2\,\mathcal{P}^2_{CHB}}{2}\right], \tag{79}$$

$$\left[\Sigma_L^{\hat{\mathcal{Z}}\hat{\mathcal{Z}}}(k^2)\right]^{div}_{\tilde{C}_{HB}} = \tilde{C}_{HB}\,\bar{v}_T^2\,\frac{g_1^2\,\mathcal{P}^2_{CHB}}{2}, \tag{80}$$

$$\left[\Sigma^{\hat{\mathcal{Z}}\hat{x}}(k^2)\right]^{div}_{\tilde{C}_{HB}} = -i\,\tilde{C}_{HB}\,\frac{\bar{v}_T\,g_1^2(\xi + 3)(3\,g_1^2 + 5\,g_2^2)}{\sqrt{g_1^2 + g_2^2}\,128\,\pi^2\,\epsilon}, \tag{81}$$

$$\left[\Sigma_T^{\hat{\mathcal{W}}^{\pm}\hat{\mathcal{W}}^{\mp}}(k^2)\right]_{\tilde{C}_{HB}}^{div} = \tilde{C}_{HB}\,\bar{v}_T^2\,\frac{g_1^2\,g_2^2\,(\xi+3)}{128\pi^2\epsilon}, \tag{82}$$

$$\left[\Sigma_L^{\hat{\mathcal{W}}^{\pm}\hat{\mathcal{W}}^{\mp}}(k^2)\right]_{\tilde{C}_{HB}}^{div} = \tilde{C}_{HB}\,\bar{v}_T^2\,\frac{g_1^2\,g_2^2\,(\xi+3)}{128\,\pi^2\,\epsilon}, \tag{83}$$

$$\left[\Sigma^{\hat{\phi}^{+}\hat{\mathcal{W}}^{-}}(k^2)\right]_{\tilde{C}_{HB}}^{div} = -\left[\Sigma^{\hat{\phi}^{-}\hat{\mathcal{W}}^{+}}(k^2)\right]_{\tilde{C}_{HB}}^{div} = -\tilde{C}_{HB}\,\bar{v}_T\,\frac{g_1^2\,g_2\,(\xi+3)}{64\,\pi^2\,\epsilon}, \tag{84}$$

$$\left[\Sigma^{\hat{\chi}\hat{\chi}}(k^2)\right]_{\tilde{C}_{HB}}^{div} = \left[\Sigma^{\hat{\phi}^{+}\hat{\phi}^{-}}(k^2)\right]_{\tilde{C}_{HB}}^{div} = \tilde{C}_{HB}\,g_1^2\left[-k^2\,\frac{\xi+3}{32\pi^2\epsilon}+\bar{v}_T^2\,\mathcal{P}_{CHB}^4\right], \tag{85}$$

$$\left[T^H\right]_{\tilde{C}_{HB}}^{div} = \tilde{C}_{HB}\,g_1^2\,\bar{v}_T^3\,\mathcal{P}_{CHB}^4; \tag{86}$$

$\left[\Sigma_L^{\hat{\mathcal{A}}\hat{\mathcal{A}}}(k^2)\right]_{\tilde{C}_{HB}}^{div}$ and $\left[\Sigma_L^{\hat{\mathcal{A}}\hat{\mathcal{Z}}}(k^2)\right]_{\tilde{C}_{HB}}^{div}$ are exactly vanishing in the BFM, consistent with the BFM Ward identities. Conversely $\left[\Sigma_T^{\hat{\mathcal{A}}\hat{\mathcal{A}}}(k^2)\right]_{\tilde{C}_{HB}}^{div}$ and $\left[\Sigma_T^{\hat{\mathcal{A}}\hat{\mathcal{A}}}(k^2)\right]_{\tilde{C}_{HB}}^{div}$ are proportional to $k^2$, and only vanish as $k^2 \to 0$. This is also consistent with the SMEFT BFM Ward identities.

The remaining Ward identities are maintained in a more intricate and interesting fashion. For example

$$
\begin{aligned}
-i\bar{M}_{\mathcal{Z}}\Sigma^{\hat{\mathcal{Z}}\hat{\chi}}(k^2) &= -i\,\frac{\sqrt{g_1^2+g_2^2}\,\bar{v}_T}{2}\left[\Sigma^{\hat{\mathcal{Z}}\hat{\chi}}(k^2)\right]_{\tilde{C}_{HB}}^{div} -i\,\frac{g_1^2\,\tilde{C}_{HB}\,\bar{v}_T}{2\sqrt{g_1^2+g_2^2}}\left[\Sigma^{\hat{\mathcal{Z}}\hat{\chi}}(k^2)\right]_{SM}^{div} \\
&= -\tilde{C}_{HB}\,\bar{v}_T^2\,(\xi+3)\left[\frac{g_1^2\,(3\,g_1^2+5\,g_2^2)}{256\,\pi^2\,\epsilon}+\frac{g_1^2\,(g_1^2+3\,g_2^2)}{256\pi^2\epsilon}\right] \\
&= -\tilde{C}_{HB}\,\bar{v}_T^2\,(\xi+3)\,\frac{g_1^2(g_1^2+2g_2^2)}{64\,\pi^2\,\epsilon}, \tag{87}
\end{aligned}
$$

which exactly cancels $\left[\Sigma_L^{\hat{\mathcal{Z}}\hat{\mathcal{Z}}}(k^2)\right]_{\tilde{C}_{HB}}^{div}$ establishing the corresponding BFM Ward identity.

Here we have not expanded out $\bar{v}_T$, simply for compact notation. Expanding $\bar{v}_T$ out in terms of the SM vev and corrections does not change the Ward identity for this operator. The manner in which the Ward identities are maintained in the SMEFT involves a nontrivial combination of the appearance of the SMEFT geometric Lagrangian parameters in the Ward identities, in conjunction with the direct evaluation of the one loop diagrams in the BFM. In the later, one must expand out the dependence on corresponding Wilson coefficient in the geometric SMEFT pole masses diagram by diagram.

Similarly, the following $\mathcal{Z}$ identity has the individual contributions

$$k^2\left[\Sigma^{\hat{\mathcal{Z}}\hat{\chi}}(k^2)\right]_{\tilde{C}_{HB}}^{div} = -i\,\tilde{C}_{HB}\,k^2\,\frac{\bar{v}_T\,g_1^2\,(\xi+3)\,(3\,g_1^2+5\,g_2^2)}{\sqrt{g_1^2+g_2^2}\,128\,\pi^2\,\epsilon}, \tag{88}$$

$$
\begin{aligned}
-i\bar{M}_{\mathcal{Z}}\left[\Sigma^{\hat{\chi}\hat{\chi}}(k^2)\right]^{div} &= -i\,\frac{\sqrt{g_1^2+g_2^2}\,\bar{v}_T}{2}\left[\Sigma^{\hat{\chi}\hat{\chi}}(k^2)\right]_{\tilde{C}_{HB}}^{div} -i\,\frac{g_1^2\,\tilde{C}_{HB}\,\bar{v}_T}{2\sqrt{g_1^2+g_2^2}}\left[\Sigma^{\hat{\chi}\hat{\chi}}(k^2)\right]_{SM}^{div} \\
&= i\,\tilde{C}_{HB}\,k^2\,\frac{\bar{v}_T\,g_1^2\,(\xi+3)\,(3\,g_1^2+5\,g_2^2)}{\sqrt{g_1^2+g_2^2}\,128\,\pi^2\,\epsilon} -i\,\frac{\tilde{C}_{HB}}{2}\,g_1^2\,\bar{v}_T^3\,\sqrt{g_1^2+g_2^2}\,\mathcal{P}_{HB}^4 \\
&\quad -i\,\frac{\tilde{C}_{HB}\,g_1^2}{2\sqrt{g_1^2+g_2^2}}\left[T^H\right]_{SM}^{div}, \tag{89}
\end{aligned}
$$

$$i\,\frac{\bar{g}_{\mathcal{Z}}}{2}T^H = i\,\frac{\tilde{C}_{HB}}{2}\,g_1^2\,\bar{v}_T^3\,\sqrt{g_1^2+g_2^2}\,\mathcal{P}_{HB}^4 +i\,\frac{\tilde{C}_{HB}\,g_1^2}{2\sqrt{g_1^2+g_2^2}}\left[T^H\right]_{SM}^{div}, \tag{90}$$

that combine to satisfy the corresponding Ward Identity.

The charged field Ward identies are satisfied directly for this operator, as

$$\left[\Sigma^{\hat{\mathcal{W}}^+\hat{\mathcal{W}}^-}(k^2)\right]_{\tilde{C}_{HB}}^{div} + \frac{g_2\,\bar{v}_T}{2}\left[\Sigma^{\hat{\phi}^+\hat{\mathcal{W}}^-}(k^2)\right]_{\tilde{C}_{HB}}^{div} = 0 \tag{91}$$

and

$$k^2\left[\Sigma^{\hat{\mathcal{W}}^+\hat{\phi}^-}(k^2)\right]_{\tilde{C}_{HB}}^{div} + \frac{g_2\,\bar{v}_T}{2}\left[\Sigma^{\hat{\phi}^-\hat{\phi}^+}(k^2)\right]_{\tilde{C}_{HB}}^{div} - \frac{g_2}{2}\left[T^H\right]_{\tilde{C}_{HB}}^{div} = 0. \tag{92}$$

### 4.3.2  Operator $Q_{HW}$

Defining the combinations of coupling which occur frequently for this operator as

$$\mathcal{P}_{CHW}^1 = \frac{((g_1^4+3\,g_2^4)\xi+4\,g_1^2\,g_2^2(\xi-7)+8\,(g_1^2+g_2^2)\lambda)}{32\pi^2\epsilon}, \tag{93}$$

$$\mathcal{P}_{CHW}^2 = \frac{(3+\xi)(2\,g_1^2+3\,g_2^2)}{32\pi^2\epsilon}, \tag{94}$$

$$\mathcal{P}_{CHW}^3 = \frac{5\,g_1^2-37g_2^2}{48\pi^2\epsilon}, \tag{95}$$

$$\mathcal{P}_{CHW}^4 = \frac{(9g_1^2+27g_2^2)+12\,\lambda\,\xi}{128\pi^2\epsilon}, \tag{96}$$

the two point functions in the SMEFT are

$$\left[\Sigma_T^{\hat{\mathcal{A}}\hat{\mathcal{A}}}(k^2)\right]_{\tilde{C}_{HW}}^{div} = \tilde{C}_{HW}\,k^2\,\frac{g_1^2\,\mathcal{P}_{CHW}^1}{(g_1^2+g_2^2)^2}, \tag{97}$$

$$\left[\Sigma_L^{\hat{\mathcal{A}}\hat{\mathcal{A}}}(k^2)\right]_{\tilde{C}_{HW}}^{div} = 0, \tag{98}$$

$$\left[\Sigma_T^{\hat{\mathcal{A}}\hat{\mathcal{Z}}}(k^2)\right]_{\tilde{C}_{HW}}^{div} = \tilde{C}_{HW}\,k^2\left[\frac{g_1\,g_2\,\mathcal{P}_{CHW}^1}{(g_1^2+g_2^2)^2}+\frac{g_1\,g_2\,\mathcal{P}_{CHB}^3}{2\,(g_1^2+g_2^2)}\right], \tag{99}$$

$$\left[\Sigma_L^{\hat{\mathcal{A}}\hat{\mathcal{Z}}}(k^2)\right]_{\tilde{C}_{HW}}^{div} = 0, \tag{100}$$

$$\left[\Sigma_T^{\hat{\mathcal{Z}}\hat{\mathcal{Z}}}(k^2)\right]_{\tilde{C}_{HW}}^{div} = \tilde{C}_{HW}\left[k^2\,\frac{g_2^2\,\mathcal{P}_{CHW}^1}{(g_1^2+g_2^2)^2}+k^2\,\frac{g_2^2\,\mathcal{P}_{CHW}^3}{(g_1^2+g_2^2)}+\bar{v}_T^2\,\frac{g_2^2\,\mathcal{P}_{CHW}^2}{2}\right], \tag{101}$$

$$\left[\Sigma_L^{\hat{\mathcal{Z}}\hat{\mathcal{Z}}}(k^2)\right]_{\tilde{C}_{HW}}^{div} = \tilde{C}_{HW}\,\bar{v}_T^2\,\frac{g_2^2\,\mathcal{P}_{CHW}^2}{2}, \tag{102}$$

$$\left[\Sigma^{\hat{\mathcal{Z}}\hat{\chi}}(k^2)\right]_{\tilde{C}_{HW}}^{div} = -i\,\tilde{C}_{HW}\,\frac{\bar{v}_T\,g_2^2\,(\xi+3)(7\,g_1^2+9\,g_2^2)}{\sqrt{g_1^2+g_2^2}\,128\,\pi^2\,\epsilon}, \tag{103}$$

$$\left[\Sigma_T^{\hat{\mathcal{W}}^\pm\hat{\mathcal{W}}^\mp}(k^2)\right]_{\tilde{C}_{HW}}^{div} = \tilde{C}_{HW}\left[k^2\,\frac{\mathcal{P}_{CHW}^1}{g_1^2+g_2^2}+k^2\,g_2^2\,\frac{\mathcal{P}_{CHW}^3}{g_1^2+g_2^2}+\bar{v}_T^2\,\frac{(g_1^2+6g_2^2)\,g_2^2\,(\xi+3)}{128\pi^2\epsilon}\right], \tag{104}$$

$$\left[\Sigma_L^{\hat{\mathcal{W}}^\pm\hat{\mathcal{W}}^\mp}(k^2)\right]_{\tilde{C}_{HW}}^{div} = \tilde{C}_{HW}\,\bar{v}_T^2\,\frac{(g_1^2+6g_2^2)\,g_2^2\,(\xi+3)}{128\pi^2\epsilon}, \tag{105}$$

$$\left[\Sigma^{\hat{\phi}^+\hat{\mathcal{W}}^-}(k^2)\right]_{\tilde{C}_{HW}}^{div} = -\left[\Sigma^{\hat{\phi}^-\hat{\mathcal{W}}^+}(k^2)\right]_{\tilde{C}_{HW}}^{div} = -\tilde{C}_{HW}\,\bar{v}_T\,\frac{(g_1^2+9g_2^2)\,g_2\,(\xi+3)}{128\,\pi^2\,\epsilon}, \tag{106}$$

$$\left[\Sigma^{\hat{\chi}\hat{\chi}}(k^2)\right]_{\tilde{C}_{HW}}^{div} = \left[\Sigma^{\hat{\phi}^+\hat{\phi}^-}(k^2)\right]_{\tilde{C}_{HW}}^{div} = \tilde{C}_{HW}\,g_2^2\left[-3\,k^2\,\frac{\xi+3}{32\pi^2\epsilon}+\bar{v}_T^2\,\mathcal{P}_{CHW}^4\right], \tag{107}$$

$$\left[T^H\right]_{\tilde{C}_{HW}}^{div} = \tilde{C}_{HW}\,g_2^2\,\bar{v}_T^3\,\mathcal{P}_{CHW}^4; \tag{108}$$

$\left[\Sigma_L^{\hat{\mathcal{A}}\hat{\mathcal{A}}}(k^2)\right]_{\tilde{C}_{HW}}^{div} = \left[\Sigma_L^{\hat{\mathcal{A}}\hat{\mathcal{Z}}}(k^2)\right]_{\tilde{C}_{HW}}^{div} = 0$ and $\left[\Sigma_T^{\hat{\mathcal{A}}\hat{\mathcal{A}}}(k^2)\right]_{\tilde{C}_{HW}}$ $\left[\Sigma_T^{\hat{\mathcal{A}}\hat{\mathcal{A}}}(k^2)\right]_{\tilde{C}_{HW}}$ have the same dependence on $k^2$ as in the case of $\tilde{C}_{HB}$. The corresponding SMEFT BFM Ward identities are satisfied in the same manner. Further, we find

$$
\begin{aligned}
-i\bar{M}_{\mathcal{Z}}\Sigma^{\hat{\mathcal{Z}}\hat{\chi}}(k^2) &= -i\frac{\sqrt{g_1^2+g_2^2}\,\bar{v}_T}{2}\left[\Sigma^{\hat{\mathcal{Z}}\hat{\chi}}(k^2)\right]_{\tilde{C}_{HW}}^{div} - i\frac{g_2^2\,\tilde{C}_{HW}\,\bar{v}_T}{2\sqrt{g_1^2+g_2^2}}\left[\Sigma^{\hat{\mathcal{Z}}\hat{\chi}}(k^2)\right]_{SM}^{div} \\
&= -\tilde{C}_{HW}\,\bar{v}_T^2(\xi+3)\left[\frac{g_2^2(7g_1^2+9g_2^2)}{256\pi^2\epsilon} + \frac{g_2^2(g_1^2+3g_2^2)}{256\pi^2\epsilon}\right] \\
&= -\tilde{C}_{HW}\,\bar{v}_T^2(\xi+3)\frac{g_2^2(2g_1^2+3g_2^2)}{64\pi^2\epsilon},
\end{aligned}
\tag{109}
$$

which exactly cancels $\left[\Sigma_L^{\hat{\mathcal{Z}}\hat{\mathcal{Z}}}(k^2)\right]_{\tilde{C}_{HW}}^{div}$.

In the case of $\tilde{C}_{HW}$, the remaining $\mathcal{Z}$ identity has the individual contributions

$$
k^2\left[\Sigma^{\hat{\mathcal{Z}}\hat{\chi}}(k^2)\right]_{\tilde{C}_{HW}}^{div} = -i\tilde{C}_{HW}\,k^2\frac{\bar{v}_T\,g_2^2(\xi+3)(7g_1^2+9g_2^2)}{\sqrt{g_1^2+g_2^2}\,128\pi^2\epsilon},
\tag{110}
$$

$$
\begin{aligned}
-i\bar{M}_{\mathcal{Z}}\left[\Sigma^{\hat{\chi}\hat{\chi}}(k^2)\right]^{div} &= -i\frac{\sqrt{g_1^2+g_2^2}\,\bar{v}_T}{2}\left[\Sigma^{\hat{\chi}\hat{\chi}}(k^2)\right]_{\tilde{C}_{HW}}^{div} - i\frac{g_2^2\,\tilde{C}_{HW}\,\bar{v}_T}{2\sqrt{g_1^2+g_2^2}}\left[\Sigma^{\hat{\chi}\hat{\chi}}(k^2)\right]_{SM}^{div} \\
&= i\tilde{C}_{HW}\,k^2\frac{\bar{v}_T\,g_2^2(\xi+3)(7g_1^2+9g_2^2)}{\sqrt{g_1^2+g_2^2}\,128\pi^2\epsilon} - i\frac{\tilde{C}_{HW}}{2}g_1^2\,\bar{v}_T^3\sqrt{g_1^2+g_2^2}\,\mathcal{P}_{HW}^4 \\
&\quad - i\frac{\tilde{C}_{HW}\,g_2^2}{2\sqrt{g_1^2+g_2^2}}\left[T^H\right]_{SM}^{div},
\end{aligned}
\tag{111}
$$

$$
i\frac{\bar{g}_{\mathcal{Z}}}{2}T^H = i\frac{\tilde{C}_{HW}}{2}g_2^2\,\bar{v}_T^3\sqrt{g_1^2+g_2^2}\,\mathcal{P}_{HW}^4 + i\frac{\tilde{C}_{HW}\,g_2^2}{2\sqrt{g_1^2+g_2^2}}\left[T^H\right]_{SM}^{div},
\tag{112}
$$

that combine to satisfy the corresponding Ward Identity.

A charged field Ward identities is satisfied directly, as

$$
\left[\Sigma_L^{\hat{\mathcal{W}}^+\hat{\mathcal{W}}^-}(k^2)\right]_{\tilde{C}_{HW}}^{div} + \frac{g_2\,\bar{v}_T}{2}\left[\Sigma^{\hat{\mathcal{W}}^-\hat{\phi}^+}(k^2)\right]_{\tilde{C}_{HW}}^{div} + \frac{g_2\bar{v}_T}{2}\tilde{C}_{HW}\left[\Sigma^{\hat{\mathcal{W}}^-\hat{\phi}^+}(k^2)\right]_{SM}^{div} = 0,
\tag{113}
$$

the remaining identity also requires the redefinition of the $\mathcal{W}$ mass into the geoSMEFT mass to be established as

$$
k^2\left[\Sigma^{\hat{\mathcal{W}}^+\hat{\phi}^-}(k^2)\right]_{\tilde{C}_{HW}}^{div} = \tilde{C}_{HW}\,k^2\,\bar{v}_T\frac{(g_1^2+9g_2^2)g_2(\xi+3)}{128\pi^2\epsilon},
\tag{114}
$$

$$
\begin{aligned}
\bar{M}_W\left[\Sigma^{\hat{\phi}^-\hat{\phi}^+}(k^2)\right] &= \frac{g_2\,\bar{v}_T}{2}\left[\Sigma^{\hat{\phi}^-\hat{\phi}^+}(k^2)\right]_{\tilde{C}_{HW}}^{div} + \frac{g_2\,\bar{v}_T}{2}\tilde{C}_{HW}\left[\Sigma^{\hat{\phi}^-\hat{\phi}^+}(k^2)\right]_{SM}^{div} \\
&= \frac{g_2}{2}\tilde{C}_{HW}\left[T^H\right]_{SM}^{div} + \frac{g_2}{2}\left[T^H\right]_{\tilde{C}_{HW}}^{div} - \tilde{C}_{HW}\,k^2\,\bar{v}_T\frac{(g_1^2+9g_2^2)g_2(\xi+3)}{128\pi^2\epsilon},
\end{aligned}
$$

$$
-\frac{\bar{g}_2}{2}\left[T^H\right] = -\frac{g_2}{2}\tilde{C}_{HW}\left[T^H\right]_{SM}^{div} - \frac{g_2}{2}\left[T^H\right]_{\tilde{C}_{HW}}^{div}.
\tag{115}
$$

Figure 1: Two point function diagrams evaluated in the SMEFT. In each diagram, all possible operator insertions are implied in the one and two point functions. Here long dashed lines are scalar fields, including Goldstone boson fields, and short dashed lines are ghost fields.

### 4.3.3 Operator $Q_{HWB}$

The Wilson coefficient of the operator $Q_{HWB}$ modifies the Weinberg angle of the SM into the appropriate rotation to mass eigenstates in the SMEFT, given in Eqn. (6). The same Wilson coefficient shifts the definition of the $\mathcal{Z}$ mass in $\bar{M}_{\mathcal{Z}}$, modifies $g_Z$ to $\bar{g}_Z$ etc. The various contributions to the BFM Ward identities combine in the following (somewhat intricate) fashion. Again defining combinations of coupling which occur frequently as

$$\mathcal{P}^1_{CHWB} = \frac{((g_1^4 + 3g_2^4)\xi + 12g_2^4 + 4g_1^2 g_2^2(\xi - 4) + 8(g_1^2 + g_2^2)\lambda)}{32\pi^2\epsilon}, \tag{116}$$

$$\mathcal{P}^2_{CHWB} = \frac{(3+\xi)(g_1^2 + 2g_2^2)}{32\pi^2\epsilon}, \tag{117}$$

$$\mathcal{P}^3_{CHWB} = \frac{g_1^2 - 3g_2^2}{32\pi^2\epsilon}, \tag{118}$$

$$\mathcal{P}^4_{CHWB} = \frac{9(g_1^2 + g_2^2) + 4\lambda\xi}{128\pi^2\epsilon}, \tag{119}$$

the two point functions in the SMEFT are

$$\left[\Sigma_T^{\hat{\mathcal{A}}\hat{\mathcal{A}}}(k^2)\right]^{div}_{\tilde{C}_{HWB}} = -\tilde{C}_{HWB}\, k^2\, \frac{g_1 g_2\, \mathcal{P}^1_{CHWB}}{(g_1^2 + g_2^2)^2}, \tag{120}$$

$$\left[\Sigma_L^{\hat{\mathcal{A}}\hat{\mathcal{A}}}(k^2)\right]^{div}_{\tilde{C}_{HWB}} = 0, \tag{121}$$

$$\left[\Sigma_T^{\hat{\mathcal{A}}\hat{\mathcal{Z}}}(k^2)\right]^{div}_{\tilde{C}_{HWB}} = \tilde{C}_{HWB}\, k^2\left[-\frac{(g_2^2 - g_1^2)\mathcal{P}^1_{CHWB}}{2(g_1^2 + g_2^2)^2} + \mathcal{P}^3_{CHWB}\right], \tag{122}$$

$$\left[\Sigma_L^{\hat{\mathcal{A}}\hat{\mathcal{Z}}}(k^2)\right]^{div}_{\tilde{C}_{HWB}} = 0, \tag{123}$$

$$\left[\Sigma_T^{\hat{\mathcal{Z}}\hat{\mathcal{Z}}}(k^2)\right]^{div}_{\tilde{C}_{HWB}} = \tilde{C}_{HWB}\, g_1 g_2\left[+k^2\frac{\mathcal{P}^1_{CHWB}}{(g_1^2 + g_2^2)^2} + \frac{k^2}{4\pi^2\epsilon} + \bar{v}_T^2\frac{\mathcal{P}^2_{CHWB}}{2}\right], \tag{124}$$

$$\left[\Sigma_L^{\hat{\mathcal{Z}}\hat{\mathcal{Z}}}(k^2)\right]^{div}_{\tilde{C}_{HWB}} = \tilde{C}_{HWB}\, \bar{v}_T^2\frac{g_1 g_2\, \mathcal{P}^2_{CHWB}}{2}, \tag{125}$$

$$\left[\Sigma^{\hat{\mathcal{Z}}\hat{\chi}}(k^2)\right]^{div}_{\tilde{C}_{HWB}} = -i\,\tilde{C}_{HWB}\, \frac{\bar{v}_T\, g_1 g_2(\xi+3)(3g_1^2 + 5g_2^2)}{\sqrt{g_1^2 + g_2^2}\, 128\pi^2\epsilon}, \tag{126}$$

$$\left[\Sigma_T^{\hat{\mathcal{W}}^\pm \hat{\mathcal{W}}^\mp}(k^2)\right]^{div}_{\tilde{C}_{HWB}} = \tilde{C}_{HWB}\, g_1 g_2\left[\frac{k^2}{16\pi^2\epsilon} + \frac{\bar{v}_T^2 g_2^2(\xi+3)}{128\pi^2\epsilon}\right], \tag{127}$$

$$\left[\Sigma_L^{\hat{\mathcal{W}}^\pm \hat{\mathcal{W}}^\mp}(k^2)\right]^{div}_{\tilde{C}_{HWB}} = \tilde{C}_{HWB}\, \bar{v}_T^2\frac{g_1 g_2^3(\xi+3)}{128\pi^2\epsilon}, \tag{128}$$

$$\left[\Sigma^{\hat{\phi}^+ \hat{\mathcal{W}}^-}(k^2)\right]^{div}_{\tilde{C}_{HWB}} = -\left[\Sigma^{\hat{\phi}^- \hat{\mathcal{W}}^+}(k^2)\right]^{div}_{\tilde{C}_{HWB}} = -\tilde{C}_{HWB}\, \bar{v}_T\frac{g_1 g_2^2(\xi+3)}{64\pi^2\epsilon}, \tag{129}$$

$$\left[\Sigma^{\hat{\chi}\hat{\chi}}(k^2)\right]^{div}_{\tilde{C}_{HWB}} = \left[\Sigma^{\hat{\phi}^+ \hat{\phi}^-}(k^2)\right]^{div}_{\tilde{C}_{HWB}} = \tilde{C}_{HWB}\, g_1 g_2\left[-k^2\frac{\xi+3}{32\pi^2\epsilon} + \bar{v}_T^2\mathcal{P}^4_{CHWB}\right], \tag{130}$$

$$\left[T^H\right]^{div}_{\tilde{C}_{HWB}} = \tilde{C}_{HWB}\, g_1 g_2\, \bar{v}_T^3\mathcal{P}^4_{CHWB}. \tag{131}$$

Once again

$$\left[\Sigma_L^{\hat{\mathcal{A}}\hat{\mathcal{A}}}(k^2)\right]^{div}_{\tilde{C}_{HWB}} = \left[\Sigma_L^{\hat{\mathcal{A}}\hat{\mathcal{Z}}}(k^2)\right]^{div}_{\tilde{C}_{HWB}} = 0, \tag{132}$$

and the fact that

$$\left[\Sigma_T^{\hat{\mathcal{A}}\hat{\mathcal{A}}}(k^2)\right]^{div}_{\tilde{C}_{HWB}} \propto k^2, \qquad\qquad \left[\Sigma_T^{\hat{\mathcal{A}}\hat{\mathcal{Z}}}(k^2)\right]^{div}_{\tilde{C}_{HWB}} \propto k^2,$$

directly establish the SMEFT BFM Ward identities involving the photon. Due to the modification of the mass parameter of the $\mathcal{Z}$ to $\bar{M}_{\mathcal{Z}}$ one finds

$$
\begin{aligned}
-i\bar{M}_{\mathcal{Z}}\Sigma^{\hat{\mathcal{Z}}\hat{\mathcal{X}}}(k^2) &= -i\frac{\sqrt{g_1^2+g_2^2}\,\bar{v}_T}{2}\left[\Sigma^{\hat{\mathcal{Z}}\hat{\mathcal{X}}}(k^2)\right]^{div}_{\tilde{C}_{HWB}} - i\frac{g_1\,g_2\,\tilde{C}_{HWB}\,\bar{v}_T}{2\sqrt{g_1^2+g_2^2}}\left[\Sigma^{\hat{\mathcal{Z}}\hat{\mathcal{X}}}(k^2)\right]^{div}_{SM}\\
&= -\tilde{C}_{HWB}\,\bar{v}_T^2\,(\xi+3)\left[\frac{g_1\,g_2\,(3\,g_1^2+5\,g_2^2)}{256\,\pi^2\,\epsilon}+\frac{g_1\,g_2\,(g_1^2+3g_2^2)}{256\pi^2\epsilon}\right]\\
&= -\tilde{C}_{HWB}\,\bar{v}_T^2\,(\xi+3)\frac{g_1g_2(g_1^2+2g_2^2)}{64\,\pi^2\,\epsilon}.
\end{aligned}
\tag{133}
$$

This combined result cancels $\left[\Sigma_L^{\hat{\mathcal{Z}}\hat{\mathcal{Z}}}(k^2)\right]^{div}_{\tilde{C}_{HWB}}$ exactly. A similar modification of $g_Z$ to $\bar{g}_Z$ in the SMEFT Ward identities in the BFM results in

$$
k^2\left[\Sigma^{\hat{\mathcal{Z}}\hat{\mathcal{X}}}(k^2)\right]^{div}_{\tilde{C}_{HWB}} = -i\,\tilde{C}_{HWB}\,k^2\frac{\bar{v}_T\,g_1\,g_2\,(\xi+3)(3\,g_1^2+5\,g_2^2)}{\sqrt{g_1^2+g_2^2}\,128\,\pi^2\,\epsilon}
\tag{134}
$$

$$
\begin{aligned}
-i\bar{M}_{\mathcal{Z}}\left[\Sigma^{\hat{\mathcal{X}}\hat{\mathcal{X}}}(k^2)\right]^{div} &= -i\frac{\sqrt{g_1^2+g_2^2}\,\bar{v}_T}{2}\left[\Sigma^{\hat{\mathcal{X}}\hat{\mathcal{X}}}(k^2)\right]^{div}_{\tilde{C}_{HWB}} - i\frac{g_1\,g_2\,\tilde{C}_{HWB}\,\bar{v}_T}{2\sqrt{g_1^2+g_2^2}}\left[\Sigma^{\hat{\mathcal{X}}\hat{\mathcal{X}}}(k^2)\right]^{div}_{SM}\\
&= i\,\tilde{C}_{HWB}k^2\frac{\bar{v}_T g_1 g_2(\xi+3)(3g_1^2+5g_2^2)}{\sqrt{g_1^2+g_2^2}128\pi^2\epsilon} - i\frac{\tilde{C}_{HWB}}{2}g_1\,g_2\,\bar{v}_T^3\sqrt{g_1^2+g_2^2}\,\mathcal{P}^4_{HWB}\\
&\quad - i\frac{\tilde{C}_{HWB}\,g_1\,g_2}{2\sqrt{g_1^2+g_2^2}}\left[T^H\right]^{div}_{SM},
\end{aligned}
\tag{135}
$$

$$
i\frac{\bar{g}_Z}{2}T^H = i\frac{\tilde{C}_{HWB}\,g_1g_2}{2}\,\bar{v}_T^3\sqrt{g_1^2+g_2^2}\,\mathcal{P}^4_{HB}+i\frac{\tilde{C}_{HWB}\,g_1\,g_2}{2\sqrt{g_1^2+g_2^2}}\left[T^H\right]^{div}_{SM}.
\tag{136}
$$

The remaining two point function Ward identities are trivially satisfied for this operator.

### 4.3.4  Operator $Q_{HD}$

For all operators in the SMEFT, a consistent analysis of the effects of an operator is essential to avoid introducing a hard breaking of a symmetry that defines the theory. The two derivative Higgs operators in $\mathcal{L}^{(6)}$ satisfy the Ward identities in a manner that involves a direct modification of tadpole contributions. Including such effects in a formulation of the SMEFT is essential, even at tree level, for the background field gauge invariance encoding unbroken but non manifest $SU(2)_L \times U(1)_Y$ symmetry of the theory to be maintained. These symmetry constraints are the Ward identities.

We define for $\tilde{C}_{HD}$ the short hand notation

$$
\mathcal{P}^1_{CHD} = \frac{2(g_1^2+3g_2^2)\xi+(9g_1^2+21g_2^2)+24\lambda}{512\pi^2\epsilon},
\tag{137}
$$

$$
\mathcal{P}^2_{CHD} = \frac{15g_1^4+30g_1^2g_2^2+9g_2^4-608\lambda^2-4\xi\lambda(g_1^2+3g_2^2)}{1024\pi^2\epsilon}.
\tag{138}
$$

The one and two point functions dependence on $\tilde{C}_{HD}$ at one loop is

$$\left[\Sigma_T^{\hat{\mathcal{A}}\hat{\mathcal{A}}}(k^2)\right]_{\tilde{C}_{HD}}^{div} = 0, \tag{139}$$

$$\left[\Sigma_L^{\hat{\mathcal{A}}\hat{\mathcal{A}}}(k^2)\right]_{\tilde{C}_{HD}}^{div} = 0, \tag{140}$$

$$\left[\Sigma_T^{\hat{\mathcal{A}}\hat{\mathcal{Z}}}(k^2)\right]_{\tilde{C}_{HD}}^{div} = -\tilde{C}_{HD}\,k^2\,\frac{g_1\,g_2}{192\,\pi^2\epsilon}, \tag{141}$$

$$\left[\Sigma_L^{\hat{\mathcal{A}}\hat{\mathcal{Z}}}(k^2)\right]_{\tilde{C}_{HD}}^{div} = 0, \tag{142}$$

$$\left[\Sigma_T^{\hat{\mathcal{Z}}\hat{\mathcal{Z}}}(k^2)\right]_{\tilde{C}_{HD}}^{div} = \tilde{C}_{HD}\left[k^2\,\frac{g_1^2}{96\pi^2\epsilon} + \bar{v}_T^2\,(g_1^2+g_2^2)\mathcal{P}_{CHD}^1\right], \tag{143}$$

$$\left[\Sigma_L^{\hat{\mathcal{Z}}\hat{\mathcal{Z}}}(k^2)\right]_{\tilde{C}_{HD}}^{div} = \tilde{C}_{HD}\,\bar{v}_T^2\,(g_1^2+g_2^2)\mathcal{P}_{CHD}^1, \tag{144}$$

$$\left[\Sigma^{\hat{\mathcal{Z}}\hat{\chi}}(k^2)\right]_{\tilde{C}_{HD}}^{div} = -i\tilde{C}_{HD}\,\bar{v}_T\,\sqrt{g_1^2+g_2^2}\,\frac{3\,(g_1^2+3g_2^2)\xi+15g_1^2+33g_2^2+48\lambda}{512\,\pi^2\,\epsilon}, \tag{145}$$

$$\left[\Sigma_T^{\hat{\mathcal{W}}^{\pm}\hat{\mathcal{W}}^{\mp}}(k^2)\right]_{\tilde{C}_{HD}}^{div} = \left[\Sigma_L^{\hat{\mathcal{W}}^{\pm}\hat{\mathcal{W}}^{\mp}}(k^2)\right]_{\tilde{C}_{HD}}^{div} = -3\,\tilde{C}_{HD}\,g_2^2\,\frac{\bar{v}_T^2\,(g_2^2-g_1^2)}{256\,\pi^2\,\epsilon}, \tag{146}$$

$$\left[\Sigma^{\hat{\phi}^+\hat{\mathcal{W}}^-}(k^2)\right]_{\tilde{C}_{HD}}^{div} = -\left[\Sigma^{\hat{\phi}^-\hat{\mathcal{W}}^+}(k^2)\right]_{\tilde{C}_{HD}}^{div} = 3\,\tilde{C}_{HD}\,\bar{v}_T\,\frac{g_2\,(g_2^2-g_1^2)}{128\,\pi^2\,\epsilon}, \tag{147}$$

$$\begin{aligned}\left[\Sigma^{\hat{\chi}\hat{\chi}}(k^2)\right]_{\tilde{C}_{HD}}^{div} &= -k^2\,\tilde{C}_{HD}\,\frac{(g_1^2+3g_2^2)\xi+6(g_1^2+2g_2^2)+24\lambda}{128\pi^2\epsilon} \\ &+ \bar{v}_T^2\,\tilde{C}_{HD}\,\frac{3g_1^2\,(g_1^2+2g_2^2)-2\lambda\,(g_1^2+3g_2^2)\xi-176\lambda^2}{256\pi^2\epsilon},\end{aligned} \tag{148}$$

$$\left[\Sigma^{\hat{\phi}^+\hat{\phi}^-}(k^2)\right]_{\tilde{C}_{HD}}^{div} = k^2\,\tilde{C}_{HD}\,\frac{3\,(g_2^2-g_1^2)}{64\,\pi^2\,\epsilon} + \bar{v}_T^2\,\tilde{C}_{HD}\,\frac{9\,(g_1^2+g_2^2)^2-256\lambda^2}{512\pi^2\epsilon}, \tag{149}$$

$$\left[T^H\right]_{\tilde{C}_{HD}}^{div} = \tilde{C}_{HD}\,\bar{v}_T^3\,\mathcal{P}_{CHD}^2. \tag{150}$$

The photon Ward identities are trivially satisfied for this operator. As $\left[\Sigma_T^{\hat{\mathcal{A}}\hat{\mathcal{Z}}}(k^2)\right]_{\tilde{C}_{HD}}^{div} \propto k^2$ the remaining identity for $\Sigma^{\hat{\mathcal{A}}\hat{\mathcal{Z}}}$ directly follows. Further, $\bar{M}_{\mathcal{Z}}$ is modified by $\tilde{C}_{HD}$ in the geoSMEFT, and one finds the expected relationship

$$\begin{aligned}-i\bar{M}_{\mathcal{Z}}\Sigma^{\hat{\mathcal{Z}}\hat{\chi}}(k^2) &= -i\,\frac{\sqrt{g_1^2+g_2^2}\,\bar{v}_T}{2}\left[\Sigma^{\hat{\mathcal{Z}}\hat{\chi}}(k^2)\right]_{\tilde{C}_{HD}}^{div} - i\sqrt{g_1^2+g_2^2}\frac{\tilde{C}_{HD}\,\bar{v}_T}{8}\left[\Sigma^{\hat{\mathcal{Z}}\hat{\chi}}(k^2)\right]_{SM}^{div} \\ &= -\tilde{C}_{HD}\,\bar{v}_T^2\,(g_1^2+g_2^2)\mathcal{P}_{CHD}^1,\end{aligned} \tag{151}$$

leading to the cancelation of $\left[\Sigma_L^{\hat{\mathcal{Z}}\hat{\mathcal{Z}}}(k^2)\right]_{\tilde{C}_{HD}}^{div}$.

The remaining $\mathcal{Z}$ identity has individual contributions

$$\begin{aligned}-i\bar{M}_{\mathcal{Z}}\left[\Sigma^{\hat{\chi}\hat{\chi}}(k^2)\right]^{div} &= -i\,\frac{\sqrt{g_1^2+g_2^2}\,\bar{v}_T}{2}\left[\Sigma^{\hat{\chi}\hat{\chi}}(k^2)\right]_{\tilde{C}_{HD}}^{div} - i\sqrt{g_1^2+g_2^2}\frac{\tilde{C}_{HD}\,\bar{v}_T}{8}\left[\Sigma^{\hat{\chi}\hat{\chi}}(k^2)\right]_{SM}^{div} \\ &= -i\,\frac{\sqrt{g_1^2+g_2^2}}{2}\left[T^H\right]_{\tilde{C}_{HD}}^{div} - k^2\left[\Sigma^{\hat{\mathcal{Z}}\hat{\chi}}(k^2)\right]_{\tilde{C}_{HD}}^{div},\end{aligned} \tag{152}$$

$$i\,\frac{\bar{g}_Z}{2}T^H = i\,\frac{\sqrt{g_1^2+g_2^2}}{2}\left[T^H\right]_{\tilde{C}_{HD}}^{div}, \tag{153}$$

that combine to satisfy the corresponding Ward Identity. The BFM Ward identity: $\Sigma_L^{\hat{\mathcal{W}}^+\hat{\mathcal{W}}^-} + \bar{M}_W\Sigma^{\hat{\mathcal{W}}^-\hat{\phi}^+} = 0$, is directly satisfied for this operator. More interesting is the modified Tadpole

contribution in the identity

$$0 = k^2 \Sigma^{\hat{\mathcal{W}}^+ \hat{\phi}^-} + \bar{M}_W \, \Sigma^{\hat{\phi}^- \hat{\phi}^+} - \frac{\bar{g}_2}{2} T^H \left( 1 + \frac{\tilde{C}_{HD}}{4} \right). \tag{154}$$

The individual terms of this Ward identity, dependent on $\tilde{C}_{HD}$ expand out as

$$k^2 \Sigma^{\hat{\mathcal{W}}^+ \hat{\phi}^-} = k^2 \left[ \Sigma^{\hat{\mathcal{W}}^+ \hat{\phi}^-} \right]^{div}_{\tilde{C}_{HD}} = -3 k^2 \, \bar{v}_T \, \tilde{C}_{HD} \, \frac{g_2 (g_2^2 - g_1^2)}{128 \, \pi^2 \epsilon}, \tag{155}$$

$$\bar{M}_W \, \Sigma^{\hat{\phi}^- \hat{\phi}^+} = \bar{v}_T \, \tilde{C}_{HD} \, g_2 \left[ 3 k^2 \, \frac{(g_2^2 - g_1^2)}{128 \, \pi^2 \epsilon} + \bar{v}_T^2 \, \frac{9 (g_1^2 + g_2^2)^2 - 256 \lambda^2}{1024 \pi^2 \epsilon} \right], \tag{156}$$

$$-\frac{\bar{g}_2}{2} T^H \left( 1 + \frac{\tilde{C}_{HD}}{4} \right) = -\frac{g_2}{2} \left[ T^H \right]^{div}_{\tilde{C}_{HD}} - \frac{g_2}{8} \tilde{C}_{HD} \left[ T^H \right]^{div}_{SM}, \tag{157}$$

and the Ward identity is satisfied as

$$\tilde{C}_{HD} \, \frac{9 (g_1^2 + g_2^2)^2 - 256 \lambda^2}{128 \pi^2 \epsilon} - \frac{4}{\bar{v}_T^3} \left[ T^H \right]^{div}_{\tilde{C}_{HD}} - \frac{\tilde{C}_{HD}}{\bar{v}_T^3} \left[ T^H \right]^{div}_{SM} = 0. \tag{158}$$

### 4.3.5 Operator $Q_{H\Box}$

The one and two point function dependence on $\tilde{C}_{H\Box}$ is

$$\left[ \Sigma_T^{\hat{\mathcal{A}} \hat{\mathcal{A}}}(k^2) \right]^{div}_{\tilde{C}_{H\Box}} = \left[ \Sigma_L^{\hat{\mathcal{A}} \hat{\mathcal{A}}}(k^2) \right]^{div}_{\tilde{C}_{H\Box}} = \left[ \Sigma_T^{\hat{\mathcal{A}} \hat{\mathcal{Z}}}(k^2) \right]^{div}_{\tilde{C}_{H\Box}} = \left[ \Sigma_L^{\hat{\mathcal{A}} \hat{\mathcal{Z}}}(k^2) \right]^{div}_{\tilde{C}_{H\Box}} = 0, \tag{159}$$

$$\left[ \Sigma_T^{\hat{\mathcal{Z}} \hat{\mathcal{Z}}}(k^2) \right]^{div}_{\tilde{C}_{H\Box}} = \tilde{C}_{H\Box} \frac{(g_1^2 + g_2^2)}{384 \, \pi^2 \, \epsilon} \left[ 4 k^2 + 9 \, \bar{v}_T^2 \, (g_1^2 + g_2^2) \right], \tag{160}$$

$$\left[ \Sigma_L^{\hat{\mathcal{Z}} \hat{\mathcal{Z}}}(k^2) \right]^{div}_{\tilde{C}_{H\Box}} = 3 \, \tilde{C}_{H\Box} \, \bar{v}_T^2 \, \frac{(g_1^2 + g_2^2)^2}{128 \, \pi^2 \, \epsilon}, \tag{161}$$

$$\left[ \Sigma^{\hat{\mathcal{Z}} \hat{\chi}}(k^2) \right]^{div}_{\tilde{C}_{H\Box}} = -3 i \, \tilde{C}_{H\Box} \, \bar{v}_T \, \sqrt{g_1^2 + g_2^2} \, \frac{(g_1^2 + g_2^2)}{64 \, \pi^2 \, \epsilon}, \tag{162}$$

$$\left[ \Sigma_T^{\hat{\mathcal{W}}^\pm \hat{\mathcal{W}}^\mp}(k^2) \right]^{div}_{\tilde{C}_{H\Box}} = \tilde{C}_{H\Box} \frac{g_2^2}{384 \pi^2 \epsilon} \left[ 4 k^2 + 9 \, g_2^2 \, \bar{v}_T^2 \right], \tag{163}$$

$$\left[ \Sigma_L^{\hat{\mathcal{W}}^\pm \hat{\mathcal{W}}^\mp}(k^2) \right]^{div}_{\tilde{C}_{H\Box}} = \tilde{C}_{H\Box} \frac{3 \, g_2^4 \, \bar{v}_T^2}{128 \pi^2 \epsilon}, \tag{164}$$

$$\left[ \Sigma^{\hat{\phi}^+ \hat{\mathcal{W}}^-}(k^2) \right]^{div}_{\tilde{C}_{H\Box}} = -\left[ \Sigma^{\hat{\phi}^- \hat{\mathcal{W}}^+}(k^2) \right]^{div}_{\tilde{C}_{H\Box}} = -3 \, \tilde{C}_{H\Box} \, \bar{v}_T \, \frac{g_2^3}{64 \pi^2 \epsilon}, \tag{165}$$

$$\left[ \Sigma^{\hat{\chi} \hat{\chi}}(k^2) \right]^{div}_{\tilde{C}_{H\Box}} = \tilde{C}_{H\Box} \left[ -3 k^2 \, \frac{g_1^2 + g_2^2}{32 \pi^2 \epsilon} + \bar{v}_T^2 \, \frac{64 \lambda^2}{32 \pi^2 \epsilon} \right], \tag{166}$$

$$\left[ \Sigma^{\hat{\phi}^+ \hat{\phi}^-}(k^2) \right]^{div}_{\tilde{C}_{H\Box}} = \tilde{C}_{H\Box} \left[ -3 k^2 \, \frac{g_2^2}{32 \pi^2 \epsilon} + \bar{v}_T^2 \, \frac{64 \lambda^2}{32 \pi^2 \epsilon} \right], \tag{167}$$

$$\left[ T^H \right]^{div}_{\tilde{C}_{H\Box}} = \tilde{C}_{H\Box} \, \bar{v}_T^3 \, \frac{3 \, g_1^4 + 6 \, g_1^2 \, g_2^2 + 9 \, g_2^4 + 608 \, \lambda^2 + 4 \, \lambda \, \xi \, (g_1^2 + 3 \, g_2^2)}{256 \, \pi^2 \, \epsilon}. \tag{168}$$

For $Q_{H\Box}$ the identities involving the photon are trivially satisfied. The identities without a tadpole contribution are also directly satisfied for this operator. For the identities involving a

tadpole contribution, the dependence on $\tilde{C}_{H\square}$ combines to satisfy the BFM Ward identity as

$$k^2 \left[ \Sigma^{\hat{Z}\hat{\chi}}(k^2) \right]^{div} = -3 i \, \tilde{C}_{H\square} k^2 \, \bar{v}_T \, \sqrt{g_1^2 + g_2^2} \frac{(g_1^2 + g_2^2)}{64 \, \pi^2 \, \epsilon}, \tag{169}$$

$$-i \bar{M}_{\mathcal{Z}} \left[ \Sigma^{\hat{\chi}\hat{\chi}}(k^2) \right]^{div} = -i \, \bar{v}_T \, \tilde{C}_{H\square} \sqrt{g_1^2 + g_2^2} \left[ -3 k^2 \frac{g_1^2 + g_2^2}{64\pi^2\epsilon} + \bar{v}_T^2 \frac{64 \lambda^2}{64\pi^2\epsilon} \right], \tag{170}$$

$$i \frac{\bar{g}_Z}{2} T^H (1 - \tilde{C}_{H\square}) = i \frac{\sqrt{g_1^2 + g_2^2}}{2} \left[ T^H \right]^{div}_{\tilde{C}_{H\square}} - i \frac{\sqrt{g_1^2 + g_2^2}}{2} \tilde{C}_{H\square} \left[ T^H \right]^{div}_{SM}$$

$$= i \, \tilde{C}_{H\square} \bar{v}_T^3 \sqrt{g_1^2 + g_2^2} \frac{\lambda^2}{\pi^2 \, \epsilon}, \tag{171}$$

and the individual terms in the corresponding charged field Ward identity, dependent on $\tilde{C}_{H\square}$ expand out as

$$k^2 \left[ \Sigma^{\hat{\mathcal{W}}^+ \hat{\Phi}^-} \right]^{div}_{\tilde{C}_{H\square}} = \tilde{C}_{H\square} \bar{v}_T k^2 \frac{3 g_2^3}{64\pi^2\epsilon}, \tag{172}$$

$$\bar{M}_W \left[ \Sigma^{\hat{\Phi}^- \hat{\Phi}^+} \right]^{div}_{\tilde{C}_{H\square}} = \tilde{C}_{H\square} \bar{v}_T \left[ -3 k^2 \frac{g_2^3}{64\pi^2\epsilon} + \bar{v}_T^2 \frac{64 g_2 \lambda^2}{64\pi^2\epsilon} \right], \tag{173}$$

$$-\frac{\bar{g}_2}{2} T^H \left( 1 - \tilde{C}_{H\square} \right) = -\frac{g_2}{2} \left[ T^H \right]^{div}_{\tilde{C}_{H\square}} + \frac{g_2}{2} \tilde{C}_{H\square} \left[ T^H \right]^{div}_{SM}$$

$$= -\tilde{C}_{H\square} \frac{g_2 \lambda^2 \bar{v}_T^3}{\pi^2 \epsilon}. \tag{174}$$

### 4.3.6 Operator $Q_H$

The operator $Q_H$ leads to a modification of the vacuum expectation value in the SM into that of the SMEFT. $Q_H$ also contributes directly to the Goldstone boson two point functions, and generates a tadpole term at one loop. It follows from the results in Ref. [10] that for this operator

$$\left[ \Sigma^{\hat{\Phi}^+ \hat{\Phi}^-} \right]^{div}_{\tilde{C}_H} = \left[ \Sigma^{\hat{\chi}\hat{\chi}} \right]^{div}_{\tilde{C}_H} = \bar{v}_T \left[ T^H \right]^{div}_{\tilde{C}_H}, \tag{175}$$

and we find this relationship holds as expected, with

$$\left[ \Sigma^{\hat{\Phi}^+ \hat{\Phi}^-} \right]^{div}_{\tilde{C}_H} = -\frac{3 \, \tilde{C}_H \, \bar{v}_T^2 \, (64\lambda + (g_1^2 + 3g_2^2) \xi)}{128\pi^2 \, \epsilon}. \tag{176}$$

### 4.3.7 Operator $Q_W$

The two point function dependence on $\tilde{C}_W$ is entirely transverse and is given by

$$\left[ \Sigma_L^{\hat{A}\hat{A}}(k^2) \right]^{div}_{\tilde{C}_W} = \left[ \Sigma_L^{\hat{A}\hat{Z}}(k^2) \right]^{div}_{\tilde{C}_W} = \left[ \Sigma_L^{\hat{Z}\hat{Z}}(k^2) \right]^{div}_{\tilde{C}_W} = \left[ \Sigma_L^{\hat{\mathcal{W}}^\pm \hat{\mathcal{W}}^\mp}(k^2) \right]^{div}_{\tilde{C}_W} = 0, \tag{177}$$

$$\left[ \Sigma_T^{\hat{A}\hat{A}}(k^2) \right]^{div}_{\tilde{C}_W} = -\frac{3 \, \tilde{C}_W \, g_1^2 \, g_2}{8\pi^2\epsilon(g_1^2 + g_2^2)} \left[ 3 g_2^2 - 2 \frac{k^2}{\bar{v}_T^2} \right] k^2, \tag{178}$$

$$\left[ \Sigma_T^{\hat{A}\hat{Z}}(k^2) \right]^{div}_{\tilde{C}_W} = -\frac{3 \, \tilde{C}_W \, g_1 \, g_2^2}{8\pi^2\epsilon(g_1^2 + g_2^2)} \left[ 3 g_2^2 - 2 \frac{k^2}{\bar{v}_T^2} \right] k^2, \tag{179}$$

$$\left[\Sigma_T^{\hat{\mathcal{Z}}\hat{\mathcal{Z}}}(k^2)\right]_{\tilde{C}_W}^{div} = -\frac{3\,\tilde{C}_W\,g_2^3}{8\pi^2\epsilon(g_1^2+g_2^2)}\left[3\,g_2^2-2\frac{k^2}{\bar{v}_T^2}\right]k^2\,, \tag{180}$$

$$\left[\Sigma_T^{\hat{\mathcal{W}}^{\pm}\hat{\mathcal{W}}^{\mp}}(k^2)\right]_{\tilde{C}_W}^{div} = -\frac{3\,\tilde{C}_W\,g_2}{8\pi^2\epsilon}\left[3\,g_2^2-2\frac{k^2}{\bar{v}_T^2}\right]k^2\,, \tag{181}$$

$$\left[\Sigma^{\hat{\mathcal{Z}}\hat{\chi}}(k^2)\right]_{\tilde{C}_W}^{div} = \left[\Sigma^{\hat{\phi}^+\hat{\mathcal{W}}^-}(k^2)\right]_{\tilde{C}_W}^{div} = \left[\Sigma^{\hat{\chi}\hat{\chi}}(k^2)\right]_{\tilde{C}_W}^{div} = \left[\Sigma^{\hat{\phi}^+\hat{\phi}^-}(k^2)\right]_{\tilde{C}_W}^{div} = 0\,, \tag{182}$$

$$\left[T^H\right]_{\tilde{C}_W}^{div} = 0\,. \tag{183}$$

As the contributions from this operator come about due to field strengths, which limits the Helicity connections, the results are purely transverse, and also proportional to $k^2$. The overall coupling dependence also directly follows from rotating the fields to mass eigenstates. For this operator, the SMEFT Ward identities are directly satisfied.

## 4.4 SMEFT results; Fermion loops

### 4.4.1 Operator $Q_{HB}$

$$\left[\Sigma_T^{\hat{\mathcal{A}}\hat{\mathcal{A}}}(k^2)\right]_{\tilde{C}_{HB}}^{div} = \tilde{C}_{HB}\frac{g_1^2\,g_2^4}{(g_1^2+g_2^2)^2}\frac{64}{9}\frac{k^2}{16\pi^2\epsilon}n\,, \tag{184}$$

$$\left[\Sigma_L^{\hat{\mathcal{A}}\hat{\mathcal{A}}}(k^2)\right]_{\tilde{C}_{HB}}^{div} = 0\,, \tag{185}$$

$$\left[\Sigma_T^{\hat{\mathcal{A}}\hat{\mathcal{Z}}}(k^2)\right]_{\tilde{C}_{HB}}^{div} = -\tilde{C}_{HB}\frac{g_1\,g_2}{(g_1^2+g_2^2)^2}\frac{4}{9}\left(5g_1^4+18g_1^2\,g_2^2-3g_2^4\right)\frac{k^2}{16\pi^2\epsilon}n\,, \tag{186}$$

$$\left[\Sigma_L^{\hat{\mathcal{A}}\hat{\mathcal{Z}}}(k^2)\right]_{\tilde{C}_{HB}}^{div} = 0\,, \tag{187}$$

$$\left[\Sigma_T^{\hat{\mathcal{Z}}\hat{\mathcal{Z}}}(k^2)\right]_{\tilde{C}_{HB}}^{div} = -\tilde{C}_{HB}\sum_{\psi}N_C^{\psi}\,m_{\psi}^2\frac{g_1^2}{16\pi^2\epsilon}+\left(\frac{8\,\tilde{C}_{HB}}{9}\frac{k^2\,n}{16\pi^2\epsilon}\right)\frac{g_1^2\,(5g_1^4+10g_1^2\,g_2^2-3g_2^4)}{(g_1^2+g_2^2)^2} \tag{188}$$

$$\left[\Sigma_L^{\hat{\mathcal{Z}}\hat{\mathcal{Z}}}(k^2)\right]_{\tilde{C}_{HB}}^{div} = -\tilde{C}_{HB}\sum_{\psi}\frac{N_C^{\psi}\,m_{\psi}^2\,g_1^2}{16\pi^2\epsilon}\,, \tag{189}$$

$$\left[\Sigma_T^{\hat{\mathcal{W}}^+\hat{\mathcal{W}}^-}(k^2)\right]_{\tilde{C}_{HB}}^{div} = \left[\Sigma_L^{\hat{\mathcal{W}}^+\hat{\mathcal{W}}^-}(k^2)\right]_{\tilde{C}_{HB}}^{div} = \left[\Sigma^{\hat{\phi}^+\hat{\phi}^-}(k^2)\right]_{\tilde{C}_{HB}}^{div} = \left[\Sigma^{\hat{\phi}^+\hat{\mathcal{W}}^-}(k^2)\right]_{\tilde{C}_{HB}}^{div} = 0\,, \tag{190}$$

$$\left[\Sigma^{\hat{\mathcal{Z}}\hat{\chi}}(k^2)\right]_{\tilde{C}_{HB}}^{div} = i\,\tilde{C}_{HB}\frac{g_1^2\,\bar{v}_T}{32\pi^2\epsilon\,\sqrt{g_1^2+g_2^2}}\sum_{\psi}N_C^{\psi}Y_{\psi}^2\,, \tag{191}$$

$$\left[\Sigma^{\hat{\phi}^+\hat{\mathcal{W}}^-}(k^2)\right]_{\tilde{C}_{HB}}^{div} = \left[\Sigma^{\hat{\chi}\hat{\chi}}(k^2)\right]_{\tilde{C}_{HB}}^{div} = \left[T^H\right]_{\tilde{C}_{HB}}^{div} = 0\,. \tag{192}$$

Most of the BFM Ward identities are trivially satisfied. These contributions come from rescaling of SM results to the two point functions through fermion loops. An interesting case is the $\mathcal{Z}$ Ward identity where the geometric $\mathcal{Z}$ mass dependence on this Wilson coefficient plays a role

$$\Sigma_L^{\hat{\mathcal{Z}}\hat{\mathcal{Z}}}-i\bar{M}_{\mathcal{Z}}\Sigma^{\hat{\mathcal{Z}}\hat{\chi}} = \left[\Sigma_L^{\hat{\mathcal{Z}}\hat{\mathcal{Z}}}\right]_{\tilde{C}_{HB}}^{div}-\frac{i\,\tilde{C}_{HB}\,g_1^2\,\bar{v}_T}{2\sqrt{g_1^2+g_2^2}}\left[\Sigma^{\hat{\mathcal{Z}}\hat{\chi}}\right]_{SM}^{div}-i\frac{\sqrt{g_1^2+g_2^2}\,\bar{v}_T}{2}\left[\Sigma^{\hat{\mathcal{Z}}\hat{\chi}}\right]_{\tilde{C}_{HB}}^{div} = 0\,. \tag{193}$$

### 4.4.2  Operator $Q_{HW}$

$$\left[\Sigma_T^{\hat{\mathcal{A}}\hat{\mathcal{A}}}(k^2)\right]_{\tilde{C}_{HW}}^{div} = \tilde{C}_{HW}\frac{g_1^4\, g_2^2}{(g_1^2+g_2^2)^2}\frac{64}{9}\frac{k^2}{16\pi^2\epsilon}\,n\,, \tag{194}$$

$$\left[\Sigma_L^{\hat{\mathcal{A}}\hat{\mathcal{A}}}(k^2)\right]_{\tilde{C}_{HW}}^{div} = 0\,, \tag{195}$$

$$\left[\Sigma_T^{\hat{\mathcal{A}}\hat{\mathcal{Z}}}(k^2)\right]_{\tilde{C}_{HW}}^{div} = -\tilde{C}_{HW}\frac{g_1\, g_2}{(g_1^2+g_2^2)^2}\frac{4}{9}\left(5g_1^4-14g_1^2\, g_2^2-3g_2^4\right)\frac{k^2}{16\pi^2\epsilon}\,n\,, \tag{196}$$

$$\left[\Sigma_L^{\hat{\mathcal{A}}\hat{\mathcal{Z}}}(k^2)\right]_{\tilde{C}_{HW}}^{div} = 0\,, \tag{197}$$

$$\left[\Sigma_T^{\hat{\mathcal{Z}}\hat{\mathcal{Z}}}(k^2)\right]_{\tilde{C}_{HW}}^{div} = -\tilde{C}_{HW}\sum_{\psi}N_C^{\psi}\, m_{\psi}^2\frac{g_2^2}{16\pi^2\epsilon}-\left(\frac{8\,\tilde{C}_{HW}}{9}\frac{k^2\,n}{16\pi^2\epsilon}\right)\frac{g_2^2\,(5g_1^4-6g_1^2\, g_2^2-3g_2^4)}{(g_1^2+g_2^2)^2} \tag{198}$$

$$\left[\Sigma_L^{\hat{\mathcal{Z}}\hat{\mathcal{Z}}}(k^2)\right]_{\tilde{C}_{HW}}^{div} = -\tilde{C}_{HW}\sum_{\psi}\frac{N_C^{\psi}\, m_{\psi}^2\, g_2^2}{16\pi^2\epsilon}\,, \tag{199}$$

$$\left[\Sigma_T^{\hat{\mathcal{W}}^+\hat{\mathcal{W}}^-}(k^2)\right]_{\tilde{C}_{HW}}^{div} = -\tilde{C}_{HW}\sum_{\psi}N_C^{\psi}\, m_{\psi}^2\frac{g_2^2}{16\pi^2\epsilon}+\left(\frac{8\,\tilde{C}_{HW}}{3}g_2^2\frac{k^2\,n}{16\pi^2\epsilon}\right)\,, \tag{200}$$

$$\left[\Sigma_L^{\hat{\mathcal{W}}^+\hat{\mathcal{W}}^-}(k^2)\right]_{\tilde{C}_{HW}}^{div} = -\tilde{C}_{HW}\sum_{\psi}\frac{N_C^{\psi}\, m_{\psi}^2\, g_2^2}{16\pi^2\epsilon}\,, \tag{201}$$

$$\left[\Sigma^{\hat{\phi}^+\hat{\phi}^-}(k^2)\right]_{\tilde{C}_{HW}}^{div} = 0\,, \tag{202}$$

$$\left[\Sigma^{\hat{\phi}^+\hat{\mathcal{W}}^-}(k^2)\right]_{\tilde{C}_{HW}}^{div} = -\left[\Sigma^{\hat{\phi}^-\hat{\mathcal{W}}^+}(k^2)\right]_{SM}^{div} = \tilde{C}_{HW}\sum_{\psi}N_C^{\psi}\, Y_{\psi}^2\,\bar{v}_T\frac{g_2}{32\,\pi^2\,\epsilon}\,, \tag{203}$$

$$\left[\Sigma^{\hat{\mathcal{Z}}\hat{\chi}}(k^2)\right]_{\tilde{C}_{HW}}^{div} = i\,\tilde{C}_{HW}\sum_{\psi}N_C^{\psi}\, Y_{\psi}^2\,\bar{v}_T\frac{g_2^2}{32\,\pi^2\,\epsilon\,\sqrt{g_1^2+g_2^2}}\,, \tag{204}$$

$$\left[\Sigma^{\hat{\chi}\hat{\chi}}(k^2)\right]_{\tilde{C}_{HW}}^{div} = \left[T^H\right]_{\tilde{C}_{HW}}^{div} = 0\,. \tag{205}$$

The BFM photon Ward identities are trivially satisfied. The remaining Ward identities we examine work out as

$$\Sigma_L^{\hat{\mathcal{Z}}\hat{\mathcal{Z}}}-i\bar{M}_{\mathcal{Z}}\Sigma^{\hat{\mathcal{Z}}\hat{\chi}} = \left[\Sigma_L^{\hat{\mathcal{Z}}\hat{\mathcal{Z}}}\right]_{\tilde{C}_{HW}}^{div}-\frac{i\,\tilde{C}_{HW}\, g_2^2\,\bar{v}_T}{2\sqrt{g_1^2+g_2^2}}\left[\Sigma^{\hat{\mathcal{Z}}\hat{\chi}}\right]_{SM}^{div}-\frac{i\sqrt{g_1^2+g_2^2}\,\bar{v}_T}{2}\left[\Sigma^{\hat{\mathcal{Z}}\hat{\chi}}\right]_{\tilde{C}_{HW}}^{div}$$
$$= 0\,, \tag{206}$$

$$k^2\Sigma^{\hat{\mathcal{Z}}\hat{\chi}}-i\bar{M}_{\mathcal{Z}}\Sigma^{\hat{\chi}\hat{\chi}}+i\frac{\bar{g}_{\mathcal{Z}}}{2}T^H = i\,\tilde{C}_{HW}\frac{g_2^2\,\bar{v}_T}{2\sqrt{g_1^2+g_2^2}}\left[k^2\sum_{\psi}\frac{N_C^{\psi}Y_{\psi}^2}{16\pi^2\epsilon}-\left[\Sigma^{\hat{\chi}\hat{\chi}}\right]_{SM}^{div}+\frac{1}{\bar{v}_T}\left[T^H\right]_{SM}^{div}\right]$$
$$= 0\,, \tag{207}$$

$$\Sigma_L^{\hat{\mathcal{W}}^{\pm}\hat{\mathcal{W}}^{\mp}}\pm\bar{M}_W\Sigma^{\hat{\mathcal{W}}^{\mp}\hat{\phi}^{\pm}} = \left[\Sigma_L^{\hat{\mathcal{W}}^{\pm}\hat{\mathcal{W}}^{\mp}}\right]_{\tilde{C}_{HW}}^{div}\pm\frac{g_2\,\bar{v}_T}{2}\left(\left[\Sigma^{\hat{\mathcal{W}}^{\mp}\hat{\phi}^{\pm}}\right]_{\tilde{C}_{HW}}^{div}\pm\tilde{C}_{HW}\left[\Sigma^{\hat{\mathcal{W}}^{\mp}\hat{\phi}^{\pm}}\right]_{SM}^{div}\right)$$
$$= 0\,, \tag{208}$$

$$k^2\Sigma^{\hat{\mathcal{W}}^{\pm}\hat{\phi}^{\mp}}\pm\bar{M}_W\Sigma^{\hat{\phi}^{\mp}\hat{\phi}^{\pm}}\mp\frac{\bar{g}_2}{2}T^H = k^2\left[\Sigma^{\hat{\mathcal{W}}^{\pm}\hat{\phi}^{\mp}}\right]_{\tilde{C}_{HW}}^{div}+\frac{\tilde{C}_{HW}\, g_2}{2}\left(\pm\bar{v}_T\left[\Sigma^{\hat{\phi}^{\mp}\hat{\phi}^{\pm}}\right]_{SM}^{div}\mp\left[T^H\right]_{SM}^{div}\right)$$
$$= 0\,. \tag{209}$$

### 4.4.3  Operator $Q_{HWB}$

$$\left[\Sigma_T^{\hat{A}\hat{A}}(k^2)\right]_{\tilde{C}_{HWB}}^{div} = -\tilde{C}_{HWB}\frac{g_1^3 g_2^3}{(g_1^2+g_2^2)^2}\frac{64}{9}\frac{k^2}{16\pi^2\epsilon}\,n,\tag{210}$$

$$\left[\Sigma_L^{\hat{A}\hat{A}}(k^2)\right]_{\tilde{C}_{HWB}}^{div} = 0,\tag{211}$$

$$\left[\Sigma_T^{\hat{A}\hat{Z}}(k^2)\right]_{\tilde{C}_{HWB}}^{div} = \tilde{C}_{HWB}\frac{g_1^2 g_2^2}{(g_1^2+g_2^2)^2}\frac{32}{9}\left(g_1^2-g_2^2\right)\frac{k^2}{16\pi^2\epsilon}\,n,\tag{212}$$

$$\left[\Sigma_L^{\hat{A}\hat{Z}}(k^2)\right]_{\tilde{C}_{HWB}}^{div} = 0,\tag{213}$$

$$\left[\Sigma_T^{\hat{Z}\hat{Z}}(k^2)\right]_{\tilde{C}_{HWB}}^{div} = -\tilde{C}_{HWB}\sum_\psi N_C^\psi\, m_\psi^2\frac{g_1 g_2}{16\pi^2\epsilon}+\left(\frac{64\,\tilde{C}_{HWB}}{9}\frac{k^2\,n}{16\pi^2\epsilon}\right)\frac{g_1^3 g_2^3}{(g_1^2+g_2^2)^2},\tag{214}$$

$$\left[\Sigma_L^{\hat{Z}\hat{Z}}(k^2)\right]_{\tilde{C}_{HWB}}^{div} = -\tilde{C}_{HWB}\sum_\psi\frac{N_C^\psi\, m_\psi^2\, g_1 g_2}{16\pi^2\epsilon},\tag{215}$$

$$\left[\Sigma_T^{\hat{W}^+\hat{W}^-}(k^2)\right]_{\tilde{C}_{HWB}}^{div} = \left[\Sigma_L^{\hat{W}^+\hat{W}^-}(k^2)\right]_{\tilde{C}_{HWB}}^{div} = 0,\tag{216}$$

$$\left[\Sigma^{\hat{\phi}^+\hat{\phi}^-}(k^2)\right]_{\tilde{C}_{HWB}}^{div} = \left[\Sigma^{\hat{\phi}^+\hat{W}^-}(k^2)\right]_{\tilde{C}_{HWB}}^{div} = 0,\tag{217}$$

$$\left[\Sigma^{\hat{Z}\hat{\chi}}(k^2)\right]_{\tilde{C}_{HWB}}^{div} = i\,\tilde{C}_{HWB}\sum_\psi N_C^\psi\, Y_\psi^2\,\bar{v}_T\frac{g_1 g_2}{32\,\pi^2\,\epsilon\,\sqrt{g_1^2+g_2^2}},\tag{218}$$

$$\left[\Sigma^{\hat{\chi}\hat{\chi}}(k^2)\right]_{\tilde{C}_{HWB}}^{div} = \left[T^H\right]_{\tilde{C}_{HWB}}^{div} = 0.\tag{219}$$

The BFM Ward identities involving the photon and charged fields are trivially satisfied. The remaining identities of interest work out as

$$\begin{aligned}\Sigma_L^{\hat{Z}\hat{Z}}-i\bar{M}_\mathcal{Z}\Sigma^{\hat{Z}\hat{\chi}} &= \left[\Sigma_L^{\hat{Z}\hat{Z}}\right]_{\tilde{C}_{HWB}}^{div}-i\frac{\sqrt{g_1^2+g_2^2}\,\bar{v}_T}{2}\left[\Sigma^{\hat{Z}\hat{\chi}}\right]_{\tilde{C}_{HWB}}^{div}\\ &\quad - i\,\tilde{C}_{HWB}\frac{g_1 g_2\bar{v}_T}{2\sqrt{g_1^2+g_2^2}}\left[\Sigma^{\hat{Z}\hat{\chi}}\right]_{SM}^{div}=0,\end{aligned}\tag{220}$$

$$\begin{aligned}k^2\Sigma^{\hat{Z}\hat{\chi}}-i\bar{M}_\mathcal{Z}\,\Sigma^{\hat{\chi}\hat{\chi}}+i\frac{\bar{g}_Z}{2}T^H &= k^2\left[\Sigma^{\hat{Z}\hat{\chi}}\right]_{\tilde{C}_{HWB}}^{div}\\ &\quad - i\,\tilde{C}_{HWB}\frac{g_1 g_2}{2\sqrt{g_1^2+g_2^2}}\left[\bar{v}_T\left[\Sigma^{\hat{\chi}\hat{\chi}}\right]_{SM}^{div}-\left[T^H\right]_{SM}^{div}\right]=0.\end{aligned}\tag{221}$$

### 4.4.4  Operator $Q_{HD}$

For this operator the non-zero divergent results for the fermion loops are

$$\left[\Sigma^{\hat{Z}\hat{\chi}}(k^2)\right]_{\tilde{C}_{HD}}^{div} = i\,\tilde{C}_{HD}\bar{v}_T\frac{\sqrt{g_1^2+g_2^2}}{128\pi^2\epsilon}\sum_\psi N_C^\psi\, Y_\psi^2,\tag{222}$$

$$\left[\Sigma^{\hat{\chi}\hat{\chi}}(k^2)\right]_{\tilde{C}_{HD}}^{div} = \frac{\tilde{C}_{HD}\bar{v}_T^2}{32\pi^2\epsilon}\sum_\psi N_C^\psi\, Y_\psi^4 - k^2\frac{\tilde{C}_{HD}}{32\pi^2\epsilon}\sum_\psi N_C^\psi\, Y_\psi^2,\tag{223}$$

$$\left[T^H\right]_{\tilde{C}_{HD}}^{div} = -\frac{\tilde{C}_{HD}}{4}\left[T^H\right]_{SM}^{div}.\tag{224}$$

Only the Ward identities involving the Tadpole contributions are non trivial for this Wilson coefficient dependence, and these results combine with the SM divergent terms from fermion loops to exactly satisfy the Ward identities.

### 4.4.5 Operator $Q_{H\Box}$

The fermion loops are simple for this operator, with only the Tadpole being non-vanishing when considering divergent terms at one loop

$$\left[T^H\right]^{div}_{\tilde{C}_{H\Box}} = \tilde{C}_{H\Box}\left[T^H\right]^{div}_{SM}, \tag{225}$$

so that

$$T^H\left(1 - \tilde{C}_{H\Box}\right) = 0. \tag{226}$$

### 4.4.6 Class 5 operators: $Q_{\psi H}$

Class five operators (in the Warsaw basis, see Table 1) can act as mass insertions and also lead to direct vertex corrections emitting goldstone bosons. In addition, a four point interaction is present that is not present in the SM which contributes to two point functions through a closed fermion loop, as shown in Fig. 1. We define the mass eigenstate Wilson coefficients

$$\tilde{C}'_{\psi H}_{pr} = \mathcal{U}^\dagger(\psi, L)\,\tilde{C}_{\psi H}\,\mathcal{U}(\psi, R), \tag{227}$$

with the rotation between mass (primed) and weak eigenstates

$$\psi_{L/R} = \mathcal{U}(\psi, L/R)\psi'_{L/R}, \tag{228}$$

where the fermion sum is over $\psi = \{u, d, \ell\}$ and $p, r$ sums over mass eigenstate flavors. The contributions to the one and two point functions are

$$\left[\Sigma_T^{\hat{\mathcal{A}}\hat{\mathcal{A}}}(k^2)\right]^{div}_{\tilde{C}_{\psi H}} = \left[\Sigma_L^{\hat{\mathcal{A}}\hat{\mathcal{A}}}(k^2)\right]^{div}_{\tilde{C}_{\psi H}} = \left[\Sigma_T^{\hat{\mathcal{A}}\hat{\mathcal{Z}}}(k^2)\right]^{div}_{\tilde{C}_{\psi H}} = \left[\Sigma_L^{\hat{\mathcal{A}}\hat{\mathcal{Z}}}(k^2)\right]^{div}_{\tilde{C}_{\psi H}} = 0, \tag{229}$$

$$\left[\Sigma_T^{\hat{\mathcal{W}}^+\hat{\mathcal{W}}^-}(k^2)\right]^{div}_{\tilde{C}_{\psi H}} = \left[\Sigma_L^{\hat{\mathcal{W}}^+\hat{\mathcal{W}}^-}(k^2)\right]^{div}_{\tilde{C}_{\psi H}} = \sum_\psi \frac{N_C^\psi\,\bar{v}_T^2\,g_2^2}{64\pi^2\epsilon}\,Y_\psi_{rr}\,\tilde{C}'_{\psi H}_{rr}, \tag{230}$$

$$\left[\Sigma_T^{\hat{\mathcal{Z}}\hat{\mathcal{Z}}}(k^2)\right]^{div}_{\tilde{C}_{\psi H}} = \left[\Sigma_L^{\hat{\mathcal{Z}}\hat{\mathcal{Z}}}(k^2)\right]^{div}_{\tilde{C}_{\psi H}} = \sum_\psi \frac{N_C^\psi\,\bar{v}_T^2\,(g_1^2 + g_2^2)}{64\pi^2\epsilon}\,Y_\psi_{pp}\,\tilde{C}'_{\psi H}_{pp}, \tag{231}$$

$$\left[\Sigma^{\hat{\mathcal{Z}}\hat{\chi}}(k^2)\right]^{div}_{\tilde{C}_{\psi H}} = -i\sum_\psi \frac{N_C^\psi\,\bar{v}_T\,\sqrt{g_1^2 + g_2^2}}{32\pi^2\epsilon}\,Y_\psi_{pp}\,\tilde{C}'_{\psi H}_{pp}, \tag{232}$$

$$-\left[\Sigma^{\hat{\phi}^+\hat{\mathcal{W}}^-}(k^2)\right]^{div}_{\tilde{C}_{\psi H}} = \left[\Sigma^{\hat{\phi}^-\hat{\mathcal{W}}^+}(k^2)\right]^{div}_{\tilde{C}_{\psi H}} = \sum_\psi \frac{N_C^\psi\,\bar{v}_T\,g_2}{32\pi^2\epsilon}\,Y_\psi_{rr}\,\tilde{C}'_{\psi H}_{rr}, \tag{233}$$

$$\left[\Sigma^{\hat{\chi}\hat{\chi}}(k^2)\right]^{div}_{\tilde{C}_{\psi H}} = -\sum_\psi \frac{k^2\,N_C^\psi}{16\pi^2\epsilon}\,Y_\psi_{pp}\,\tilde{C}'_{\psi H}_{pp} + \sum_\psi \frac{3\,N_C^\psi\,\bar{v}_T^2}{16\pi^2\epsilon}\,Y_\psi^3_{pp}\,\tilde{C}'_{\psi H}_{pp}, \tag{234}$$

$$\left[\Sigma^{\hat{\phi}^+\hat{\phi}^-}(k^2)\right]^{div}_{\tilde{C}_{\psi H}} = -\sum_\psi \frac{k^2\,N_C^\psi}{16\pi^2\epsilon}\,Y_\psi_{pp}\,\tilde{C}'_{\psi H}_{pp} + \sum_\psi \frac{3\,N_C^\psi\,\bar{v}_T^2}{16\pi^2\epsilon}\,Y_\psi^3_{pp}\,\tilde{C}'_{\psi H}_{pp}, \tag{235}$$

$$\left[T^H\right]^{div}_{\tilde{C}_{\psi H}} = \sum_\psi \frac{3\,N_C^\psi\,\bar{v}_T^3}{16\pi^2\epsilon}\,Y_\psi^3_{pp}\,\tilde{C}'_{\psi H}_{pp}. \tag{236}$$

The Ward identities are satisfied in the same manner as those in the SM involving fermion loops.

### 4.4.7 Class 6 operators: $Q_{eB}$, $Q_{dB}$, $Q_{uB}$

Class six operators (see Table 1) only act as vertex corrections. We define the mass eigenstate Wilson coefficients

$$\tilde{C}'_{\psi B}_{pr} = \mathcal{U}^\dagger(\psi, L) \tilde{C}_{\psi B} \mathcal{U}(\psi, R), \tag{237}$$

and find

$$\left[\Sigma_L^{\hat{\mathcal{A}}\hat{\mathcal{A}}}(k^2)\right]^{div}_{\tilde{C}'_{\psi B}} = \left[\Sigma_L^{\hat{\mathcal{A}}\hat{\mathcal{Z}}}(k^2)\right]^{div}_{\tilde{C}'_{\psi B}} = \left[\Sigma_L^{\hat{\mathcal{W}}^+\hat{\mathcal{W}}^-}(k^2)\right]^{div}_{\tilde{C}'_{\psi B}} = \left[\Sigma_L^{\hat{\mathcal{Z}}\hat{\mathcal{Z}}}(k^2)\right]^{div}_{\tilde{C}'_{\psi B}} = 0, \tag{238}$$

$$\left[\Sigma_T^{\hat{\mathcal{A}}\hat{\mathcal{A}}}(k^2)\right]^{div}_{\tilde{C}'_{\psi B}} = -\frac{g_1 g_2^2 k^2}{(g_1^2 + g_2^2)4\pi^2\epsilon}\left[Q_e Y_e \tilde{C}'_{eB} + Q_d N_c Y_d \tilde{C}'_{dB} + Q_u N_c Y_u \tilde{C}'_{uB} + h.c.\right], \tag{239}$$

$$\left[\Sigma_T^{\hat{\mathcal{A}}\hat{\mathcal{Z}}}(k^2)\right]^{div}_{\tilde{C}'_{\psi B}} = \frac{g_2 k^2}{(g_1^2 + g_2^2)32\pi^2\epsilon}\left[(g_2^2 - 7g_1^2)Y_e \tilde{C}'_{eB} + (13g_1^2 - 3g_2^2)Y_u \tilde{C}'_{uB} + h.c.\right]$$
$$+ \frac{g_2 k^2}{(g_1^2 + g_2^2)32\pi^2\epsilon}\left[(3g_2^2 - 5g_1^2)Y_d \tilde{C}'_{dB} + h.c.\right], \tag{240}$$

$$\left[\Sigma_T^{\hat{\mathcal{Z}}\hat{\mathcal{Z}}}(k^2)\right]^{div}_{\tilde{C}'_{\psi B}} = \frac{g_1 k^2}{(g_1^2 + g_2^2)16\pi^2\epsilon}\left[(3g_1^2 - g_2^2)Y_e \tilde{C}'_{eB} + \frac{(3g_2^2 - 5g_1^2)N_C}{3}Y_u \tilde{C}'_{uB} + h.c.\right]$$
$$+ \frac{g_1 k^2}{(g_1^2 + g_2^2)16\pi^2\epsilon}\left[\frac{(g_1^2 - 3g_2^2)N_C}{3}Y_d \tilde{C}'_{dB} + h.c.\right], \tag{241}$$

$$\left[\Sigma_T^{\hat{\mathcal{W}}^+\hat{\mathcal{W}}^-}(k^2)\right]^{div}_{\tilde{C}'_{\psi B}} = \left[\Sigma^{\hat{\mathcal{Z}}\hat{\chi}}(k^2)\right]^{div}_{\tilde{C}'_{\psi B}} = -\left[\Sigma^{\hat{\phi}^+\hat{\mathcal{W}}^-}(k^2)\right]^{div}_{\tilde{C}'_{\psi B}} = \left[\Sigma^{\hat{\phi}^-\hat{\mathcal{W}}^+}(k^2)\right]^{div}_{\tilde{C}'_{\psi B}} = 0, \tag{242}$$

$$\left[\Sigma^{\hat{\chi}\hat{\chi}}(k^2)\right]^{div}_{\tilde{C}'_{\psi B}} = \left[\Sigma^{\hat{\phi}^+\hat{\phi}^-}(k^2)\right]^{div}_{\tilde{C}'_{\psi B}} = \left[T^H\right]^{div}_{\tilde{C}'_{\psi B}} = 0. \tag{243}$$

Here $Q_\psi = (-1, -1/3, 2/3)$ for $\psi = (e, d, u)$. As the non-vanishing divergent results are purely transverse, the SMEFT Ward identities are trivially satisfied. A subset of these results can be checked against the literature, and they do agree with Ref. [35].

### 4.4.8 Class 6 operators: $Q_{eW}$, $Q_{dW}$, $Q_{uW}$

We define mass eigenstate Wilson coefficients in the same manner for this operator class and find

$$\left[\Sigma_L^{\hat{\mathcal{A}}\hat{\mathcal{A}}}(k^2)\right]^{div}_{\tilde{C}'_{\psi W}} = \left[\Sigma_L^{\hat{\mathcal{A}}\hat{\mathcal{Z}}}(k^2)\right]^{div}_{\tilde{C}'_{\psi W}} = \left[\Sigma_L^{\hat{\mathcal{W}}^+\hat{\mathcal{W}}^-}(k^2)\right]^{div}_{\tilde{C}'_{\psi W}} = \left[\Sigma_L^{\hat{\mathcal{Z}}\hat{\mathcal{Z}}}(k^2)\right]^{div}_{\tilde{C}'_{\psi W}} = 0 \tag{244}$$

$$\left[\Sigma_T^{\hat{\mathcal{A}}\hat{\mathcal{A}}}(k^2)\right]^{div}_{\tilde{C}'_{\psi W}} = \frac{g_1^2 g_2 k^2}{(g_1^2 + g_2^2)4\pi^2\epsilon}\left[Q_e Y_e \tilde{C}'_{eW} + N_c Q_d Y_d \tilde{C}'_{dW} + N_c Q_u Y_u \tilde{C}'_{uW} + h.c.\right], \tag{245}$$

$$\left[\Sigma_T^{\hat{\mathcal{A}}\hat{\mathcal{Z}}}(k^2)\right]^{div}_{\tilde{C}'_{\psi W}} = \frac{g_1 k^2}{(g_1^2 + g_2^2)32\pi^2\epsilon}\left[(3g_1^2 - 5g_2^2)Y_e \tilde{C}'_{eW} + (5g_1^2 - 11g_2^2)Y_u \tilde{C}'_{uW} + h.c.\right]$$
$$+ \frac{g_2 k^2}{(g_1^2 + g_2^2)32\pi^2\epsilon}\left[(g_1^2 - 7g_2^2)Y_d \tilde{C}'_{dW} + h.c.\right], \tag{246}$$

$$\left[\Sigma_T^{\hat{\mathcal{Z}}\hat{\mathcal{Z}}}(k^2)\right]^{div}_{\tilde{C}'_{\psi W}} = \frac{g_2 k^2}{(g_1^2 + g_2^2)16\pi^2\epsilon}\left[(3g_1^2 - g_2^2)Y_e \tilde{C}'_{eW} + (5g_1^2 - 3g_2^2)Y_u \tilde{C}'_{uW} + h.c.\right]$$
$$+ \frac{g_2 k^2}{(g_1^2 + g_2^2)16\pi^2\epsilon}\left[(g_1^2 - 3g_2^2)Y_d \tilde{C}'_{dW} + h.c.\right], \tag{247}$$

$$\left[\Sigma_T^{\hat{\mathcal{W}}^+\hat{\mathcal{W}}^-}(k^2)\right]_{\tilde{C}'_{\psi W}}^{div} = -\frac{g_2\,k^2}{(g_1^2+g_2^2)16\,\pi^2}\left[U_{PMNS}\underset{pr}{\tilde{C}'_{eW}}\underset{rq}{Y_e}\underset{pq}{}+h.c.\right] \tag{248}$$

$$-\frac{g_2\,k^2}{(g_1^2+g_2^2)16\,\pi^2}\left[N_C\,V_{CKM}\underset{pr}{\tilde{C}'_{uW}}\underset{rq}{Y_u}\underset{pq}{}+N_C\,V_{CKM}\underset{pr}{\tilde{C}'_{dW}}\underset{rq}{Y_d}\underset{pq}{}+h.c.\right],$$

$$\left[\Sigma^{\hat{\mathcal{Z}}\hat{\chi}}(k^2)\right]_{\tilde{C}'_{\psi W}}^{div} = -\left[\Sigma^{\hat{\phi}^+\hat{\mathcal{W}}^-}(k^2)\right]_{\tilde{C}'_{\psi W}}^{div} = \left[\Sigma^{\hat{\phi}^-\hat{\mathcal{W}}^+}(k^2)\right]_{\tilde{C}'_{\psi W}}^{div} = 0, \tag{249}$$

$$\left[\Sigma^{\hat{\chi}\hat{\chi}}(k^2)\right]_{\tilde{C}'_{\psi W}}^{div} = \left[\Sigma^{\hat{\phi}^+\hat{\phi}^-}(k^2)\right]_{\tilde{C}'_{\psi W}}^{div} = \left[T^H\right]_{\tilde{C}'_{\psi W}}^{div} = 0. \tag{250}$$

Once again, the non-vanishing divergent results are purely transverse, and the SMEFT Ward identities are trivially satisfied.

### 4.4.9 Class 7 operators: $Q_{He}$, $Q_{Hu}$, $Q_{Hd}$, $Q_{Hud}$

For this operator class, we define the mass eigenstate Wilson coefficients

$$\underset{pr}{\tilde{C}'_{H\psi_R}} = \mathcal{U}^\dagger(\psi,R)\,\tilde{C}_{H\psi_R}\,\mathcal{U}(\psi,R), \tag{251}$$

and note that only the flavour diagonal contributions $r = p$ contribute at one loop due to the lack of flavour changing neutral currents in tree level couplings in the SM. Directly we find

$$\left[\Sigma_T^{\hat{\mathcal{A}}\hat{\mathcal{A}}}(k^2)\right]_{\tilde{C}_{H\psi_R}}^{div} = \left[\Sigma_L^{\hat{\mathcal{A}}\hat{\mathcal{A}}}(k^2)\right]_{\tilde{C}_{H\psi_R}}^{div} = 0, \tag{252}$$

$$\left[\Sigma_T^{\hat{\mathcal{A}}\hat{\mathcal{Z}}}(k^2)\right]_{\tilde{C}_{H\psi_R}}^{div} = -\frac{g_1\,g_2\,k^2}{24\pi^2\epsilon}\left[Q_e\underset{pp}{\tilde{C}'_{He}}+N_c\,Q_u\underset{pp}{\tilde{C}'_{Hu}}+N_C\,Q_d\underset{pp}{\tilde{C}'_{Hd}}\right], \tag{253}$$

$$\left[\Sigma_L^{\hat{\mathcal{A}}\hat{\mathcal{Z}}}(k^2)\right]_{\tilde{C}_{H\psi_R}}^{div} = 0, \tag{254}$$

$$\left[\Sigma_T^{\hat{\mathcal{W}}^+\hat{\mathcal{W}}^-}(k^2)\right]_{\tilde{C}_{H\psi_R}}^{div} = -\underset{pr}{\tilde{C}'_{Hud}}\underset{pr}{V_{CKM}^\star}\frac{g_2^2\,N_C\,\bar{v}_T^2\,\underset{rr}{Y_d}\,\underset{pp}{Y_u}}{64\pi^2\epsilon}+h.c., \tag{255}$$

$$\left[\Sigma_L^{\hat{\mathcal{W}}^+\hat{\mathcal{W}}^-}(k^2)\right]_{\tilde{C}_{H\psi_R}}^{div} = -\underset{pr}{\tilde{C}'_{Hud}}\underset{pr}{V_{CKM}^\star}\frac{g_2^2\,N_C\,\bar{v}_T^2\,\underset{rr}{Y_d}\,\underset{pp}{Y_u}}{64\pi^2\epsilon}+h.c., \tag{256}$$

$$\left[\Sigma_T^{\hat{\mathcal{Z}}\hat{\mathcal{Z}}}(k^2)\right]_{\tilde{C}_{H\psi_R}}^{div} = \frac{g_1^2\,k^2}{12\pi^2\epsilon}\left[Q_e\underset{pp}{\tilde{C}'_{He}}+N_C\,Q_u\underset{pp}{\tilde{C}'_{Hu}}+Q_d\,N_C\underset{pp}{\tilde{C}'_{Hd}}\right] \tag{257}$$

$$+\frac{(g_1^2+g_2^2)\bar{v}_T^2}{16\pi^2\epsilon}\left[\underset{pp}{\tilde{C}'_{He}}\underset{pp}{Y_e^2}-N_C\underset{pp}{\tilde{C}'_{Hu}}\underset{pp}{Y_u^2}+N_C\underset{pp}{\tilde{C}'_{Hd}}\underset{pp}{Y_d^2}\right], \tag{258}$$

$$\left[\Sigma_L^{\hat{\mathcal{Z}}\hat{\mathcal{Z}}}(k^2)\right]_{\tilde{C}_{H\psi_R}}^{div} = \frac{(g_1^2+g_2^2)\bar{v}_T^2}{16\pi^2\epsilon}\left[\underset{pp}{\tilde{C}'_{He}}\underset{pp}{Y_e^2}-N_C\underset{pp}{\tilde{C}'_{Hu}}\underset{pp}{Y_u^2}+N_C\underset{pp}{\tilde{C}'_{Hd}}\underset{pp}{Y_d^2}\right], \tag{259}$$

$$\left[\Sigma^{\hat{\mathcal{Z}}\hat{\chi}}(k^2)\right]^{div}_{\tilde{C}_{H\psi_R}} = -i\frac{\sqrt{g_1^2+g_2^2}\,\bar{v}_T}{8\pi^2\epsilon}\left[\tilde{C}'_{He}\underset{pp}{Y_e^2}\underset{pp}{} - N_C\,\tilde{C}'_{Hu}\underset{pp}{Y_u^2}\underset{pp}{} + N_C\,\tilde{C}'_{Hd}\underset{pp}{Y_d^2}\underset{pp}{}\right], \tag{260}$$

$$\left[\Sigma^{\hat{\chi}\hat{\chi}}(k^2)\right]^{div}_{\tilde{C}_{H\psi_R}} = -\left[\tilde{C}'_{He}\underset{pp}{Y_e^2}\underset{pp}{} - N_C^q\,\tilde{C}'_{Hu}\underset{pp}{Y_u^2}\underset{pp}{} + N_C\,\tilde{C}'_{Hd}\underset{pp}{Y_d^2}\underset{pp}{}\right]\frac{k^2}{4\pi^2\epsilon}, \tag{261}$$

$$\left[\Sigma^{\hat{\phi}^-\hat{\mathcal{W}}^+}(k^2)\right]^{div}_{\tilde{C}_{H\psi_R}} = -\left[\Sigma^{\hat{\phi}^+\hat{\mathcal{W}}^-}(k^2)\right]^{div}_{\tilde{C}_{H\psi_R}} = -\tilde{C}'_{Hud}\underset{pr}{V^\star_{CKM}}\underset{pr}{}\frac{g_2\,N_C\,\bar{v}_T\,Y^d_{rr}\,Y^u_{pp}}{32\pi^2\epsilon} + h.c., \tag{262}$$

$$\left[\Sigma^{\hat{\phi}^+\hat{\phi}^-}(k^2)\right]^{div}_{\tilde{C}_{H\psi_R}} = \tilde{C}'_{Hud}\underset{pr}{V^\star_{CKM}}\underset{pr}{}\frac{N_C\,Y_d\,Y_u}{16\pi^2\epsilon}\underset{rr}{}\underset{pp}{} + h.c., \tag{263}$$

$$\left[T^H\right]^{div}_{\tilde{C}_{H\psi_R}} = 0. \tag{264}$$

These results directly satisfy the corresponding SMEFT Ward identities.

### 4.4.10  Class 7 operators: $Q_{H\ell}^{(1)}, Q_{Hq}^{(1)}$

For the left handed fermion operators in this class, we similarly define the mass eigenstate Wilson coefficients

$$\tilde{C}'^{(1,3)}_{\substack{H\psi_L\\pr}} = \mathcal{U}^\dagger(\psi,L)\,\tilde{C}^{(1,3)}_{H\psi_L}\,\mathcal{U}(\psi,L). \tag{265}$$

Again, only the flavour diagonal contributions $r = p$ contribute at one loop due to the lack of flavour changing neutral currents in tree level couplings in the SM. We find

$$\left[\Sigma_T^{\hat{\mathcal{A}}\hat{\mathcal{A}}}(k^2)\right]^{div}_{\tilde{C}'^{(1)}_{H\psi_L}} = \left[\Sigma_L^{\hat{\mathcal{A}}\hat{\mathcal{A}}}(k^2)\right]^{div}_{\tilde{C}'^{(1)}_{H\psi_L}} = 0, \tag{266}$$

$$\left[\Sigma_T^{\hat{\mathcal{A}}\hat{\mathcal{Z}}}(k^2)\right]^{div}_{\tilde{C}'^{(1)}_{H\psi_L}} = \left[\tilde{C}'^{(1)}_{H\ell}\underset{pp}{} - \frac{N_c}{3}\tilde{C}'^{(1)}_{Hq}\underset{pp}{}\right]\frac{g_1\,g_2\,k^2}{48\pi^2\epsilon}, \tag{267}$$

$$\left[\Sigma_L^{\hat{\mathcal{A}}\hat{\mathcal{Z}}}(k^2)\right]^{div}_{\tilde{C}'^{(1)}_{H\psi_L}} = 0, \tag{268}$$

$$\left[\Sigma_T^{\hat{\mathcal{W}}^+\hat{\mathcal{W}}^-}(k^2)\right]^{div}_{\tilde{C}'^{(1)}_{H\psi_L}} = \left[\Sigma_L^{\hat{\mathcal{W}}^+\hat{\mathcal{W}}^-}(k^2)\right]^{div}_{\tilde{C}'^{(1)}_{H\psi_L}} = 0, \tag{269}$$

$$\left[\Sigma_T^{\hat{\mathcal{Z}}\hat{\mathcal{Z}}}(k^2)\right]^{div}_{\tilde{C}'^{(1)}_{H\psi_L}} = \left[\frac{N_C}{3}\tilde{C}'^{(1)}_{Hq}\underset{pp}{} - \tilde{C}'^{(1)}_{H\ell}\underset{pp}{}\right]\frac{g_1^2\,k^2}{24\pi^2\epsilon}$$
$$- \frac{(g_1^2+g_2^2)\bar{v}_T^2}{32\pi^2\epsilon}\left[N_C\,\tilde{C}'^{(1)}_{Hq}\underset{pp}{}(Y_d^2\underset{pp}{} - Y_u^2\underset{pp}{}) + \tilde{C}'^{(1)}_{H\ell}\underset{pp}{}Y_e^2\underset{pp}{}\right], \tag{270}$$

$$\left[\Sigma_L^{\hat{\mathcal{Z}}\hat{\mathcal{Z}}}(k^2)\right]^{div}_{\tilde{C}'^{(1)}_{H\psi_L}} = -\frac{(g_1^2+g_2^2)\bar{v}_T^2}{32\pi^2\epsilon}\left[N_C\,\tilde{C}'^{(1)}_{Hq}\underset{pp}{}(Y_d^2\underset{pp}{} - Y_u^2\underset{pp}{}) + \tilde{C}'^{(1)}_{H\ell}\underset{pp}{}Y_e^2\underset{pp}{}\right], \tag{271}$$

$$\left[\Sigma^{\hat{\mathcal{Z}}\hat{\chi}}(k^2)\right]^{div}_{\tilde{C}'^{(1)}_{H\psi_L}} = i\left[N_C\,\tilde{C}'^{(1)}_{Hq}\underset{pp}{}(Y_d^2\underset{pp}{} - Y_u^2\underset{pp}{}) + \tilde{C}'^{(1)}_{H\ell}\underset{pp}{}Y_e^2\underset{pp}{}\right]\frac{\sqrt{g_1^2+g_2^2}\,\bar{v}_T}{16\pi^2\epsilon}, \tag{272}$$

$$\left[\Sigma^{\hat{\chi}\hat{\chi}}(k^2)\right]^{div}_{\tilde{C}'^{(1)}_{H\psi_L}} = \left[N_C\,\tilde{C}'^{(1)}_{Hq}\underset{pp}{}(Y_d^2\underset{pp}{} - Y_u^2\underset{pp}{}) + \tilde{C}'^{(1)}_{H\ell}\underset{pp}{}Y_e^2\underset{pp}{}\right]\frac{k^2}{8\pi^2\epsilon}, \tag{273}$$

$$\left[\Sigma^{\hat{\phi}^-\hat{\mathcal{W}}^+}(k^2)\right]^{div}_{\tilde{C}'^{(1)}_{H\psi_L}} = \left[\Sigma^{\hat{\phi}^+\hat{\phi}^-}(k^2)\right]^{div}_{\tilde{C}'^{(1)}_{H\psi_L}} = \left[T^H\right]^{div}_{\tilde{C}'^{(1)}_{H\psi_L}} = 0. \tag{274}$$

Again the SMEFT Ward identities are directly satisfied by these expressions.

### 4.4.11 Class 7 operators: $Q_{H\ell}^{(3)}, Q_{Hq}^{(3)}$

In this case one finds

$$\left[\Sigma_T^{\hat{\mathcal{A}}\hat{\mathcal{A}}}(k^2)\right]_{\tilde{C}_{H\psi_L}^{\prime(3)}}^{div} = \left[\Sigma_L^{\hat{\mathcal{A}}\hat{\mathcal{A}}}(k^2)\right]_{\tilde{C}_{H\psi_L}^{\prime(3)}}^{div} = 0, \tag{275}$$

$$\left[\Sigma_T^{\hat{\mathcal{A}}\hat{\mathcal{Z}}}(k^2)\right]_{\tilde{C}_{H\psi_L}^{\prime(3)}}^{div} = \left[N_C\,\tilde{C}_{Hq}^{\prime(3)} + \tilde{C}_{H\ell}^{\prime(3)}\right]\frac{g_1\,g_2\,k^2}{48\pi^2\epsilon}, \tag{276}$$

$$\left[\Sigma_L^{\hat{\mathcal{A}}\hat{\mathcal{Z}}}(k^2)\right]_{\tilde{C}_{H\psi_L}^{\prime(3)}}^{div} = 0, \tag{277}$$

$$\left[\Sigma_T^{\hat{\mathcal{W}}^+\hat{\mathcal{W}}^-}(k^2)\right]_{\tilde{C}_{H\psi_L}^{\prime(3)}}^{div} = \left[N_C\,\tilde{C}_{Hq}^{\prime(3)}\,V_{CKM}^\star + \tilde{C}_{H\ell}^{\prime(3)}\,U_{PMNS}^\star + h.c.\right]\frac{g_2^2\,k^2}{48\pi^2\epsilon} \tag{278}$$

$$- \frac{g_2^2\,\bar{v}_T^2}{64\,\pi^2\,\epsilon}\left[N_C\,\tilde{C}_{Hq}^{\prime(3)}\,V_{CKM}^\star(Y_u^2 + Y_d^2) + \tilde{C}_{H\ell}^{\prime(3)}\,U_{PMNS}^\star\,Y_e^2 + h.c.\right], \tag{279}$$

$$\left[\Sigma_L^{\hat{\mathcal{W}}^+\hat{\mathcal{W}}^-}(k^2)\right]_{\tilde{C}_{H\psi_L}^{\prime(3)}}^{div} = -\frac{g_2^2\,\bar{v}_T^2}{64\,\pi^2\,\epsilon}\left[N_C\,\tilde{C}_{Hq}^{\prime(3)}\,V_{CKM}^\star(Y_u^2 + Y_d^2) + \tilde{C}_{H\ell}^{\prime(3)}\,U_{PMNS}^\star\,Y_e^2 + h.c.\right], \tag{280}$$

$$\left[\Sigma_T^{\hat{\mathcal{Z}}\hat{\mathcal{Z}}}(k^2)\right]_{\tilde{C}_{H\psi_L}^{\prime(3)}}^{div} = \left[N_C\,\tilde{C}_{Hq}^{\prime(3)} + \tilde{C}_{H\ell}^{\prime(3)}\right]\frac{g_2^2\,k^2}{24\pi^2\epsilon} \tag{281}$$

$$- \left[N_C\,\tilde{C}_{Hq}^{\prime(3)}(Y_u^2 + Y_d^2) + \tilde{C}_{H\ell}^{\prime(3)}\,Y_e^2\right]\frac{(g_1^2 + g_2^2)\,\bar{v}_T^2}{32\,\pi^2\,\epsilon}, \tag{282}$$

$$\left[\Sigma_L^{\hat{\mathcal{Z}}\hat{\mathcal{Z}}}(k^2)\right]_{\tilde{C}_{H\psi_L}^{\prime(3)}}^{div} = -\left[N_C\,\tilde{C}_{Hq}^{\prime(3)}(Y_u^2 + Y_d^2) + \tilde{C}_{H\ell}^{\prime(3)}\,Y_e^2\right]\frac{(g_1^2 + g_2^2)\,\bar{v}_T^2}{32\,\pi^2\,\epsilon}, \tag{283}$$

$$\left[\Sigma^{\hat{\mathcal{Z}}\hat{\chi}}(k^2)\right]_{\tilde{C}_{H\psi_L}^{\prime(3)}}^{div} = i\left[N_C\,\tilde{C}_{Hq}^{\prime(3)}(Y_u^2 + Y_d^2) + \tilde{C}_{H\ell}^{\prime(3)}\,Y_e^2\right]\frac{\sqrt{g_1^2 + g_2^2}\,\bar{v}_T}{16\pi^2\epsilon}, \tag{284}$$

$$\left[\Sigma^{\hat{\chi}\hat{\chi}}(k^2)\right]_{\tilde{C}_{H\psi_L}^{\prime(3)}}^{div} = \left[N_C\,\tilde{C}_{Hq}^{\prime(3)}(Y_u^2 + Y_d^2) + \tilde{C}_{H\ell}^{\prime(3)}\,Y_e^2\right]\frac{k^2}{8\pi^2\epsilon}, \tag{285}$$

$$-\left[\Sigma^{\hat{\phi}^-\hat{\mathcal{W}}^+}(k^2)\right]_{\tilde{C}_{H\psi_L}^{\prime(3)}}^{div} = \left[\Sigma^{\hat{\phi}^+\hat{\mathcal{W}}^-}(k^2)\right]_{\tilde{C}_{H\psi_L}^{\prime(3)}}^{div}$$

$$= \frac{g_2\,\bar{v}_T}{32\,\pi^2\,\epsilon}\left[N_C\,\tilde{C}_{Hq}^{\prime(3)}\,V_{CKM}^\star(Y_u^2 + Y_d^2) + \tilde{C}_{H\ell}^{\prime(3)}\,U_{PMNS}^\star\,Y_e^2 + h.c.\right], \tag{286}$$

$$\left[\Sigma^{\hat{\phi}^+\hat{\phi}^-}(k^2)\right]_{\tilde{C}_{H\psi_L}^{\prime(3)}}^{div} = \frac{k^2}{16\pi^2\epsilon}\left[N_C\,\tilde{C}_{Hq}^{\prime(3)}\,V_{CKM}^\star(Y_u^2 + Y_d^2) + \tilde{C}_{H\ell}^{\prime(3)}\,U_{PMNS}^\star\,Y_e^2 + h.c.\right], \tag{287}$$

$$\left[T^H\right]_{\tilde{C}_{H\psi_L}^{\prime(3)}}^{div} = 0. \tag{288}$$

Again, these results directly satisfy the corresponding SMEFT Ward identities.

## 5 Discussion

Theoretical consistency checks, such as the BFM Ward identities examined and validated at one loop in this work, are useful because they allow internal cross checks of theoretical calculations, and provide a means of validating numerical codes that can be used for experimental studies. This is of increased importance in the SMEFT, which is a complex field theory.

It is important to stress that the Ward identities are always modified transitioning to the SMEFT from the SM, but the nature of the changes to the identities depends on the gauge fixing procedure. If the Background Field Method is not used, then only more complicated Slavnov-Taylor [36–38] identities hold. These identities also necessarily involve modifications from the SM case due to the presence of SMEFT operators. The derivation in Ref. [9], that is expanded upon in this work, should make clear why this is necessarily the case. The identities are modified because the Lagrangian quantities on the curved background Higgs manifold's present, that the correlation functions are quantized on, and related in the Ward or Slavnov-Taylor identities, are the natural generalization of the coupling constants and masses of the SM for these field spaces.

To our knowledge, the first discussion on the need to modify these identities in the SMEFT in the literature is in Ref. [39], and this point is also consistent with discussion in Ref. [40,41], which recognizes this modification of Ward identities is present.

In the literature, one loop calculations have been done in the SMEFT within the BFM [15, 22, 35, 42–46], and also outside of the BFM [23, 40, 41, 47–53]. It is important, when comparing results, that one recognizes that radiative scheme dependence, includes differing dependence on Wilson coefficients in the two point functions. These functions differ in the BFM in the SMEFT, compared to other schemes, because the corresponding symmetry constraints encoded in the Ward identities or Slavnov-Taylor identities also differ. Scheme dependence is manifestly a very significant issue in the SMEFT when seeking to build up a global fit, which will necessarily combine many predictions produced from multiple research groups. It is important that scheme and input parameter dependence is clearly and completely specified in a one loop SMEFT calculation to aid this effort, and one should not misunderstand scheme dependence, and equate differences found in results in different schemes with error when comparing. In this work, we avoid such an elementary mistake. In any case, we stress again that in the SMEFT, in any gauge fixing approach, the Ward identities, or Slavnov-Taylor identities, necessarily differ from those in the SM.[4]

We also emphasize the appearance of the two derivative Higgs operators in the Ward identities, modifying the tadpole contributions. This is consistent with, and an explicit representation of, the discussions in Refs. [16, 54, 55]. The subtle appearance of such corrections again show the need to take the SMEFT to mass eigenstate interactions in a consistent manner.[5] A consistent treatment of the SMEFT to all orders in $\bar{v}_T/\Lambda$ [3] while preserving background field invariance leads directly to the geoSMEFT. This approach also gives an intuitive interpretation of how and why the Lagrangian parameters are modified, due to the presence of the curved Higgs field spaces modifying correlation functions.

## 6 Conclusions

In this paper we have validated Ward identities in the SMEFT at one loop, when calculating using the Background Field Method approach to gauge fixing. These results lay the groundwork for generating numerical codes to next to leading order in both the perturbative and non-perturbative expansions in the theory while using the Background Field Method in the geoSMEFT. The results also offer a clarifying demonstration on the need to carefully define SMEFT mass eigenstate interactions, to ensure that the theory is formulated in a consistent manner. Utilizing the Background Field Method is of increased utility (in the opinion of the authors of this paper) in the case of the SMEFT, as this is an effective theory including a Higgs field. Any correct formulation of the SMEFT is consistent with the assumed $SU(2)_L \times U(1)_Y$

---

[4]For an alternative point of view on these issues see Ref. [49]

[5]It is interesting to compare the treatment of such effects in this work to Ref. [56]

Table 1: The independent dimension-six operators built from Standard Model fields which conserve baryon number, as given in Ref. [2]. Four-fermion operators have been removed as they aren't relevant to this analysis. The operators are divided into seven classes: $X^3$, $H^6$, etc. Operators with +h.c. in the table heading also have hermitian conjugates, as does the $\psi^2 H^2 D$ operator $Q_{Hud}$. The subscripts $p, r$ are flavor indices. Table taken from Ref. [42].

| 1 : $X^3$ | |
|---|---|
| $Q_G$ | $f^{ABC} G_\mu^{A\nu} G_\nu^{B\rho} G_\rho^{C\mu}$ |
| $Q_{\widetilde{G}}$ | $f^{ABC} \widetilde{G}_\mu^{A\nu} G_\nu^{B\rho} G_\rho^{C\mu}$ |
| $Q_W$ | $\epsilon^{IJK} W_\mu^{I\nu} W_\nu^{J\rho} W_\rho^{K\mu}$ |
| $Q_{\widetilde{W}}$ | $\epsilon^{IJK} \widetilde{W}_\mu^{I\nu} W_\nu^{J\rho} W_\rho^{K\mu}$ |

| 2 : $H^6$ | |
|---|---|
| $Q_H$ | $(H^\dagger H)^3$ |

| 3 : $H^4 D^2$ | |
|---|---|
| $Q_{H\Box}$ | $(H^\dagger H)\Box(H^\dagger H)$ |
| $Q_{HD}$ | $\left(H^\dagger D_\mu H\right)^* \left(H^\dagger D_\mu H\right)$ |

| 4 : $X^2 H^2$ | |
|---|---|
| $Q_{HG}$ | $H^\dagger H\, G_{\mu\nu}^A G^{A\mu\nu}$ |
| $Q_{H\widetilde{G}}$ | $H^\dagger H\, \widetilde{G}_{\mu\nu}^A G^{A\mu\nu}$ |
| $Q_{HW}$ | $H^\dagger H\, W_{\mu\nu}^I W^{I\mu\nu}$ |
| $Q_{H\widetilde{W}}$ | $H^\dagger H\, \widetilde{W}_{\mu\nu}^I W^{I\mu\nu}$ |
| $Q_{HB}$ | $H^\dagger H\, B_{\mu\nu} B^{\mu\nu}$ |
| $Q_{H\widetilde{B}}$ | $H^\dagger H\, \widetilde{B}_{\mu\nu} B^{\mu\nu}$ |
| $Q_{HWB}$ | $H^\dagger \tau^I H\, W_{\mu\nu}^I B^{\mu\nu}$ |
| $Q_{H\widetilde{W}B}$ | $H^\dagger \tau^I H\, \widetilde{W}_{\mu\nu}^I B^{\mu\nu}$ |

| 5 : $\psi^2 H^3$ + h.c. | |
|---|---|
| $Q_{eH}$ | $(H^\dagger H)(\bar{l}_p e_r H)$ |
| $Q_{uH}$ | $(H^\dagger H)(\bar{q}_p u_r \widetilde{H})$ |
| $Q_{dH}$ | $(H^\dagger H)(\bar{q}_p d_r H)$ |

| 6 : $\psi^2 X H$ + h.c. | |
|---|---|
| $Q_{eW}$ | $(\bar{l}_p \sigma^{\mu\nu} e_r)\tau^I H W_{\mu\nu}^I$ |
| $Q_{eB}$ | $(\bar{l}_p \sigma^{\mu\nu} e_r) H B_{\mu\nu}$ |
| $Q_{uG}$ | $(\bar{q}_p \sigma^{\mu\nu} T^A u_r)\widetilde{H}\, G_{\mu\nu}^A$ |
| $Q_{uW}$ | $(\bar{q}_p \sigma^{\mu\nu} u_r)\tau^I \widetilde{H}\, W_{\mu\nu}^I$ |
| $Q_{uB}$ | $(\bar{q}_p \sigma^{\mu\nu} u_r)\widetilde{H}\, B_{\mu\nu}$ |
| $Q_{dG}$ | $(\bar{q}_p \sigma^{\mu\nu} T^A d_r)H\, G_{\mu\nu}^A$ |
| $Q_{dW}$ | $(\bar{q}_p \sigma^{\mu\nu} d_r)\tau^I H\, W_{\mu\nu}^I$ |
| $Q_{dB}$ | $(\bar{q}_p \sigma^{\mu\nu} d_r)H\, B_{\mu\nu}$ |

| 7 : $\psi^2 H^2 D$ | |
|---|---|
| $Q_{Hl}^{(1)}$ | $(H^\dagger i\overleftrightarrow{D}_\mu H)(\bar{l}_p \gamma^\mu l_r)$ |
| $Q_{Hl}^{(3)}$ | $(H^\dagger i\overleftrightarrow{D}_\mu^I H)(\bar{l}_p \tau^I \gamma^\mu l_r)$ |
| $Q_{He}$ | $(H^\dagger i\overleftrightarrow{D}_\mu H)(\bar{e}_p \gamma^\mu e_r)$ |
| $Q_{Hq}^{(1)}$ | $(H^\dagger i\overleftrightarrow{D}_\mu H)(\bar{q}_p \gamma^\mu q_r)$ |
| $Q_{Hq}^{(3)}$ | $(H^\dagger i\overleftrightarrow{D}_\mu^I H)(\bar{q}_p \tau^I \gamma^\mu q_r)$ |
| $Q_{Hu}$ | $(H^\dagger i\overleftrightarrow{D}_\mu H)(\bar{u}_p \gamma^\mu u_r)$ |
| $Q_{Hd}$ | $(H^\dagger i\overleftrightarrow{D}_\mu H)(\bar{d}_p \gamma^\mu d_r)$ |
| $Q_{Hud}$ + h.c. | $i(\widetilde{H}^\dagger D_\mu H)(\bar{u}_p \gamma^\mu d_r)$ |

symmetry at one loop, and this can be checked by comparing against the Ward-Takahashi or Slavnov-Taylor identities. We encourage those developing alternative formulations of the SMEFT to demonstrate the consistency of their results with the corresponding symmetry constraints classically, and at one loop, to ensure that the various approaches are all well defined.

In this work we have demonstrated that the Ward identities provide an excellent opportunity to cross check loop calculations performed in the SMEFT. In future works, this will allow for consistency checks of relevant full one-loop contributions to the effective action. For example, the full one-loop calculation of the $W$-boson propagator can be consistency checked against the full loop calculation of $\mathcal{W}$-$\phi$ mixing. The background field method will also allow for Dyson resummation of the one-loop corrections to the propagator without breaking gauge invariance [32]. To the best of the authors' knowledge, no works concerning the SMEFT have formulated or confirmed the corresponding Slavnov-Taylor identities for traditional $R_\xi$ gauge fixing. This work provides a clear foundation from which these next steps can be approached.

# Acknowledgments

We acknowledge support from the Villum Fonden, project number 00010102, and the Danish National Research Foundation through a DFF project grant. TC acknowledges funding from European Union's Horizon 2020 research and innovation programme under the Marie Sklodowska-Curie grant agreement No. 890787. We thank A. Helset and G. Passarino for discussions. MT thanks CERN for generous support during a Corona split SASS visit when much of this work was developed.

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
