# Peer review of "One loop verification of SMEFT Ward Identities"

_SciPost Physics, doi:SciPost Phys. 10, 144 (2021)_

## Round 1 · Referee Report · Anonymous (Referee 1) · 2021-3-22

Strengths
The authors verify Ward identities in the Standard Model Effective Field Theory, using the background field gauge. This remarkable technical exercise is detailed extensively.
Weaknesses
Such a verification appears rather academic and its intrinsic usefulness is unclear to me. As a consistency check, I understand it helped to validate the computer code presented in a companion paper but, on its own, does not seem to warrant publication as original research material. It does not present conceptual advances with respect to the existing literature, including works by the present authors and collaborators.
Report
My recommendation is therefore to not publish this manuscript.

---

## Round 1 · Referee Report · Anonymous (Referee 2) · 2021-5-15

Report
In this manuscript the authors check a selection of one-loop Ward
identities in the SMEFT using the background field method. Although
such calculations are already available in the literature they have
not been previously compiled in this complete fashion in a single
reference. This will be a useful resource as the precision of the data
necessitates the inclusion of one-loop SMEFT effects in future
studies.
Before publication I encourage the authors to tone down a provocative
statement made in conclusions. They state that the use of the Background Field Method is "arguably essential" in the SMEFT, as it is an effective
theory that contains a Higgs field. I disagree with this statement;
consistent one-loop calculations in the SMEFT have been made using
other approaches.
With this minor change I recommend the paper for publication.
identities in the SMEFT using the background field method. Although
such calculations are already available in the literature they have
not been previously compiled in this complete fashion in a single
reference. This will be a useful resource as the precision of the data
necessitates the inclusion of one-loop SMEFT effects in future
studies.
Before publication I encourage the authors to tone down a provocative
statement made in conclusions. They state that the use of the Background Field Method is "arguably essential" in the SMEFT, as it is an effective
theory that contains a Higgs field. I disagree with this statement;
consistent one-loop calculations in the SMEFT have been made using
other approaches.
With this minor change I recommend the paper for publication.

---

## Round 2 · Referee Report · Anonymous (Referee 2) · 2021-6-8

Report

I recommend the paper for publication. The minor change I suggested in my previous report was made by the authors.

---

## Round 2 · Author Response

We thank the editor and the reviewers for their comments and reviewing the paper.

For the second, positive reviewer’s comments we have modified the statement the reviewer pointed out as a bit strong to a more personal statement on the authors' opinion. The modified statement is:

“Utilizing the Background Field Method is of increased utility (in the opinion of the authors of this paper) in the case of the SMEFT, as this is an effective theory including a Higgs field.”

With this change we believe the second reviewer will be satisfied. We appreciate the fair mindedness of this reviewer, and the we also appreciate the editor’s inclination to suggest the paper is published.

For the first reviewer, we find little to engage with in constructive criticism on the paper. As the reviewer expresses their opinion on the paper, which of course they are entitled to, here we point out the aspects of the extensive results of the paper that inform the opinion of the authors on the value of this work. We stress these points to scientifically argue why we consider the paper a valuable and novel contribution that should be published in the literature:

1) The SMEFT calculations reported are novel. As far as we know, there is no background field method based gauge fixing demonstration of Ward identities at one loop in the published literature before this work.

2) The explicit demonstration of the BFM Ward identities we examine at one loop operator by operator clarified to the authors of the paper how the Ward identities are satisfied at one loop. One aspect of the physics that was not obvious to the authors, until this detailed calculation was done, was the effect of mass corrections in the propagators proportional to SMEFT Wilson coefficients. We found the need to expand out the mass effects in the propagators to satisfy the Ward ID, somewhat to our surprise, when performing this work. This need to expand out these contributions is mentioned in the text in sections 4.3.3 and 4.3.4 and elsewhere. This makes clear the need to expand out such contributions in SMEFT global fits.

3) The behavior of the corrections to the tadpole terms for some operators is quite novel and directly related to some issues that have been pointed out in the literature previously in alternate formulations of the SMEFT at the dimension six level. This is mentioned on in the discussion on page 5. An explicit demonstration of the behavior of the theory at one loop in these shifts to tadpole corrections is in our view a valuable and clarifying contribution to the literature.

4) Similarly, the discussion in section 5 of the paper also points out that the 2 point functions behavior is of course dependent on the gauge fixing choice of the theory. This is important to clarify in our view in the published literature, as there are comparisons across different calculational scheme choices in Ref.49 where such differences are implied to be associated with calculational error, as opposed to simply being an aspect of scheme dependence.

For all of these reasons, and primarily point 1) we consider this paper to be publishable work that will add to the literature in a useful fashion. Again we thank the reviewers and the editor for their consideration of the paper in Scipost and we hope with our response and the minor revisions the paper is now acceptable for publication in Scipost.

---

## Round 2 · List of Changes

We have modified the statement the second reviewer pointed out as too strongly worded to:
“Utilizing the Background Field Method is of increased utility (in the opinion of the authors of this paper) in the case of the SMEFT, as this is an effective theory including a Higgs field.”

---

## Editorial Decision

published